# Crystallization of piezoceramic films on glass via flash lamp annealing

Longfei Song [1,2,4], Juliette Cardoletti [1,4], Alfredo Blázquez Martínez [1,2], Andreja Benčan[3], Brigita Kmet[3], Stéphanie Girod[1], Emmanuel Defay[1] & Sebastjan Glinšek [1] ✉

Integration of thin-film oxide piezoelectrics on glass is imperative for the next generation of transparent electronics to attain sensing and actuating functions. However, their crystallization temperature (above 650 °C) is incompatible with most glasses. We developed a flash lamp process for the growth of piezoelectric lead zirconate titanate films. The process enables crystallization on various types of glasses in a few seconds only. The functional properties of these films are comparable to the films processed with standard rapid thermal annealing at 700 °C. A surface haptic device was fabricated with a 1 μm-thick film (piezoelectric $e_{33,f}$ of −5 C m$^{-2}$). Its ultrasonic surface deflection reached 1.5 μm at 60 V, sufficient for its use in surface rendering applications. This flash lamp annealing process is compatible with large glass sheets and roll-to-roll processing and has the potential to significantly expand the applications of piezoelectric devices on glass.

An important trend in next-generation large-scale electronics is the integration of functional films on glass wafers for smart windows and display screens. Among the available materials for sensors and actuators, piezoelectric oxide thin films are outstanding because of their superior electromechanical response compared to nitrides and polymers[1–5]. The two last ones are excellent materials for resonators and energy harvesters, respectively, and can be processed at low temperatures (e.g., <350 °C for AlN - and ~150 °C for polyvinyl difluoride (PVDF)-based materials). But much lower piezoelectric coefficients prevent them to replace perovskites, especially for actuator applications[6–8]. The key for the successful integration of perovskites in microelectromechanical systems (MEMS) has been efficient processing, which allows for the preparation of high-quality films in a controllable and reproducible manner. Chemical solution deposition (CSD) is among the most popular fabrication methods due to its low cost, flexibility in chemical composition and compatibility with large-scale microelectronics[9,10]. The method is continuously evolving and emerging digital printing and roll-to-roll processing technologies will enable high-speed, high-throughput and large-scale additive manufacturing[5,11].

In CSD processing the as-deposited amorphous phase is transformed into a crystalline perovskite piezoelectric phase via high-temperature annealing. The crystallisation process traditionally relies on the use of isothermal heating in tube, box, or rapid thermal annealing (RTA) furnaces (see Fig. 1a)[12]. The typical processing temperatures are above 650 °C and the total annealing time (including heating and cooling) is in the order of tens of minutes. As the strain point - the maximum temperature at which glass can be used without experiencing creep - of most commercial glasses lies below 600 °C, the development of low-temperature processing is required[13,14].

Efforts have been dedicated to lowering processing temperature, including photochemical processing[15,16], annealing under high-pressure of O$_2$/O$_3$[17], combustion synthesis[18], and laser annealing[19,20]. While the global temperature has been lowered, the price to pay is either long annealing times (often several hours) or the use of a reactive atmosphere, which limit the use of these processes in large-scale and high-throughput production. Although laser annealing allows for the fast crystallization of films, the small laser spot size (typically in the μm$^2$-mm$^2$ range) imposes the need for raster scanning of the laser beam, leading to inhomogeneities in the films[21]. The pulsed laser

[1]Materials Research and Technology Department, Luxembourg Institute of Science and Technology, 41 rue du Brill, L-4422 Belvaux, Luxembourg. [2]University of Luxembourg, 41 rue du Brill, L-4422 Belvaux, Luxembourg. [3]Electronic Ceramics Department, Jožef Stefan Institute, Jamova cesta 39, 1000 Ljubljana, Slovenia. [4]These authors contributed equally: Longfei Song, Juliette Cardoletti. ✉e-mail: sebastjan.glinsek@list.lu

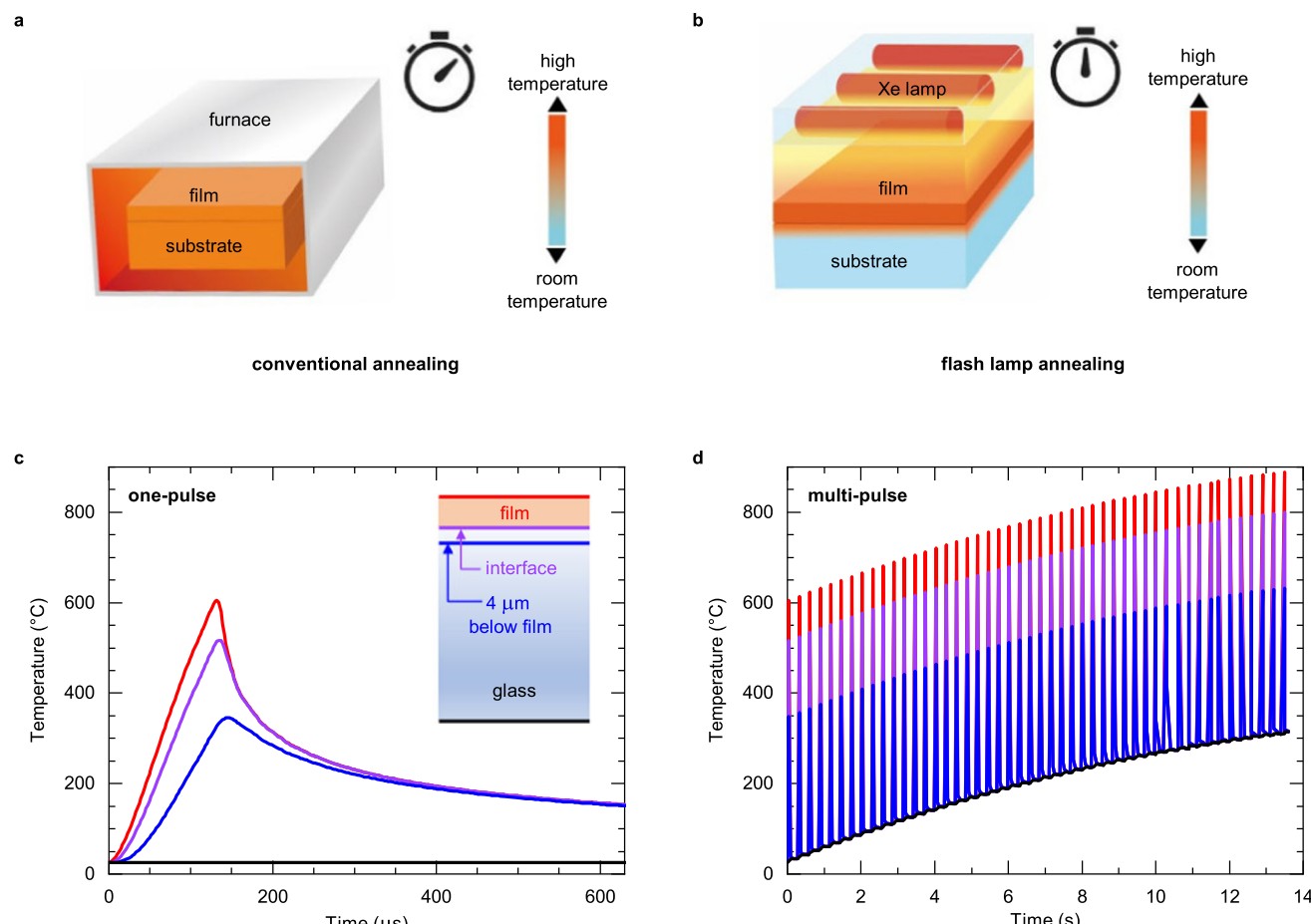

**Fig. 1 | Schematic representation of annealing processes.** Heat distribution in a film, its substrate, and processing time in the case of **a** conventional box furnace annealing and **b** flash lamp annealing. **Finite element modelling**. Temperature profiles during **c** single-pulse and **d** 50-pulse flash lamp annealing of a 170 nm-PZT/ fused silica glass stack. The red, purple, blue, and black lines correspond to the temperature profiles of the top PZT surface, the interface between the film and glass, 4 μm below this interface, and the bottom of the substrate (500 μm), respectively. Absolute temperature values are indicative only. The schematic structure of the sample is shown in the inset of **c**. The pulse duration, energy density and repetition rate are 130 μs, 3 J cm$^{-2}$, and 3.5 Hz, respectively.

deposition (PLD) method can lead to good films at 445 °C, however, low pressures (0.05 mbar) are required, and the method is difficult to scale-up[22,23]. The indirect integration on glass via the transfer process has been successful[24], however, it significantly complicates the process and adds processing steps.

Flash lamp annealing (FLA), i.e., annealing through the absorption of sub-millisecond light pulses with broad spectrum and strong intensity, enables selective annealing of the films on temperature-sensitive substrates[25] and is schematically compared to box-furnace annealing in Fig. 1b. FLA can be performed in an ambient atmosphere in a few seconds and the irradiated areas can reach several hundreds of cm². These features are compatible with high-throughput, low-cost and large-scale roll-to-roll production[26]. FLA has been successfully used for processing different functional thin films, such as conductors (e.g., indium tin oxide, metal nanoparticle films)[27,28], semiconductors (e.g., indium zinc oxide)[29,30], dielectrics (e.g., alumina, zirconia)[31,32], hybrid perovskites for absorbers in solar cells[33] and light-emitting diodes[34]. Yao et al. induced nano-crystallization of piezoelectric lead zirconate titanate on glass and polyimide substrates, however, macroscopic electromechanical properties, which are imperative for sensor and actuator applications, were not demonstrated[35].

Piezoelectrics have been recently demonstrated as efficient actuators for haptics. The technology is based on an ultrasonically vibrating surface, which can modulate forces at the interface between a finger and a vibrating plate. The so-called friction-

modulation effect is applicable in touchscreens with foreseen applications such as displays for the automotive industry, displays for visually disabled people, or sliders for control of heating/ cooling[36]. The effect depends strongly on the out-of-plane displacement, in-plane wavelength, and frequency of the standing acoustic wave, which is created on the screen. For technology commercialization, the following criteria have to be fulfilled: amplitude must be larger than 1 μm (enabling detection with a human finger and significant decrease of the friction coefficient)[37,38], wavelength must be below ~15 mm (enabling detection with a human finger)[39] and its frequency should be beyond 25 kHz (enabling silent operation)[40].

Inspired by the above results and guided by finite element modelling (FEM), we developed a process that enables macroscopic crystallization of solution-processed PbZr$_{0.53}$Ti$_{0.47}$O$_3$ (PZT) films on a wide variety of glasses. First, we demonstrate a fast process (several seconds per crystallization) on 1 μm-thick PZT thin films on fused silica, resulting in remanent polarization $P_r$, dielectric permittivity $\varepsilon_r$, dielectric losses tan$\delta$ and piezoelectric coefficient $e_{33,f}$ values of 12 μC cm$^{-2}$, 450, 5 %, and −5 C m$^{-2}$, respectively. A surface haptic device was realized on alumina-borosilicate glass (AF32, Schott), i.e., a standard substrate for semiconductor and MEMS industries. The device is based on a 1 μm-thick PZT film and its ultrasonic-range surface deflection reaches 1.5 μm at 60 V, which fulfils the requirement of a deflection of 1 μm for its commercialization in texture rendering operation[37]. Finally, we demonstrate the universality of the

FLA process by direct growth of PZT films on soda-lime, i.e., the most widespread type of glass.

Hence, in this paper, we have shown that it is possible to manufacture a functional device (a haptic transducer) based on piezoelectric perovskite thin films deposited on a glass substrate and sintered with a specific flash lamp annealing process. Moreover, we showed that a distinctive feature of the latter is that this crystallization can be performed on glass substrates that cannot withstand temperatures larger than 400 °C.

## Results

### Processes design and finite element modelling
Results of the finite element modelling are presented in Figs. 1c and d, where a 170 nm PZT/fused silica glass configuration was modelled. The absorbance of an amorphous PZT layer (~28.5 %) was estimated with a bolometer placed below the sample, considering that glass is mainly transparent in the ~300–1000 nm wavelength range of the Xe lamp utilised in this work (see Supplementary Note 1 for further information). The modelling of a 130 μs pulse with an energy density of 3 J cm$^{-2}$, indicates that the temperature at the top surface of PZT should reach 600 °C, which is sufficiently high to crystallize PZT into the piezoelectric perovskite phase[9]. Through the glass thickness, the temperature drops significantly as the process is non-adiabatic and performed in an ambient environment. This maintains the interface between the film and the substrate at a relatively low temperature, i.e., the temperature exceeds 400 °C for only 50 μs per pulse (see the purple line in Fig. 1c). 4 μm below this interface the temperature remains below 350 °C. To provide enough energy and time for complete crystallization to occur, multi-pulse annealing is necessary. 30 pulses at 3.5 Hz enable the temperature of PZT to reach 800 °C while the bulk of the glass substrates remains below 400 °C (Fig. 1d).

### Growth, Phase Composition and Microstructure
The films were deposited through a standard CSD process using 2-methoxyethanol as solvent, lead acetate and transition metals alkoxides precursors[10]. The solutions were spin-coated, dried at 130 °C and pyrolyzed at 350 °C on hot plates. The deposition-drying-pyrolysis process was repeated four times to achieve a thickness of 170 nm. The crystallization was done with flash lamp annealing. Thick films were obtained by repeating the whole process several times.

Films on fused silica were investigated first and their phase composition was analysed as a function of the number of FLA light pulses. Pulsing was performed with the same conditions as described in the previous section and in Supplementary Note 2. After 10 pulses, {100} and {110} reflections of the perovskite phase appear in the grazing incidence X-ray diffraction pattern (Fig. 2a) of 170 nm-thick films. Reflection of the non-piezoelectric pyrochlore phase (a small peak at $2\theta = 29°$), which is kinetically stabilized during annealing[41], disappears when the number of pulses is increased to 30. When the number of pulses reaches 100, the intensity of perovskite reflections increases indicating enhanced crystallinity. The phase evolution agrees with the temperature evolution predicted with FEM during multi-pulse annealing (Fig. 1d), i.e., a gradual increase of temperature stabilizes the perovskite phase. The standard $\theta$-$2\theta$ X-ray diffraction (XRD) pattern of 1 μm-thick PZT on fused silica is shown in Supplementary Fig. 2, and only perovskite peaks are present. In contrast, the perovskite phase does not form in the film annealed at 700 °C in RTA (Supplementary Fig. 3)[42]. To the best of our knowledge, without the use of a nucleation layer (or bottom electrode), successful crystallization of solution-processed PZT films on amorphous substrates has not yet been reported, showing that the developed FLA process is even excelling a typical layer-by-layer deposition of solution-processed films in standard furnaces. It also enables the growth of films in a controllable and repeatable manner.

Transmittance of the 170 nm-thick films (50 pulses) at a wavelength of 550 nm is 64 %, as shown in Fig. 2b, together with its visual appearance. Transparency is preserved in the 1 μm film on 2″ glass wafer (inset of Supplementary Fig. 4).

A cross-sectional scanning transmission electron microscope (STEM) analysis was also performed. A dark field STEM image (Fig. 2c) reveals relatively porous granular microstructure, which agrees with the scanning electron microscopy micrograph of the 1 μm-thick film shown in Supplementary Fig. 5. The high-resolution STEM image, with the corresponding Fast Fourier Transform (FFT) (inset in Fig. 2e), shows a (110) plane reflection with an interplanar spacing of 2.8 Å, which is further demonstrated in the zoomed-in STEM image (inset in Fig. 2e). This interplanar spacing value is in good agreement with the perovskite phase of PZT and is additional confirmation of its presence in the film. While the film is chemically inhomogeneous (Supplementary Fig. 6) the energy dispersive X-ray spectroscopy 2D map (Fig. 2c) shows no diffusion of Pb into the substrate (as commonly observed in conventionally processed films)[43]. It is also observed that there was partial diffusion of silicon into the PZT film, at a depth of approximately 100 nm (Fig. 2d). However, this diffusion did not result in the formation of a secondary phase that would compromise the ferroelectric and piezoelectric properties of the PZT film, as confirmed earlier in grazing incidence X-ray diffraction (GIXRD) patterns and later by electrical measurements.

### Electromechanical characterization
This section is focused on the films prepared with 50 pulses, as they show optimal ferroelectric response. Electrical properties were measured in interdigitated geometry employing surface Pt electrodes. Polarization versus electric field $P(E)$ loops are initially pinched (Supplementary Fig. 7). This is most likely caused by the presence of charged defects (such as oxygen vacancies) in the films, which at low electric fields pin ferroelectric domain walls[44]. Electric-field cycling (wake-up) enables their redistribution, which unpins the polarization. Indeed, the pinched hysteresis opens during cycling (Supplementary Fig. 7). Different numbers of wake-up cycles in different films could be due to different concentrations of defects in the films and/or their different pinning energy.

Ferroelectric and piezoelectric properties are shown in Fig. 3 and were obtained on a 1 μm-thick film on fused silica glass after 10$^4$ wake-up cycles. The maximum polarization $P_{max}$ and remanent polarization $P_r$ are 25 μC cm$^{-2}$ and 12 μC cm$^{-2}$, respectively. Its coercive field $E_c$ is 68 kV cm$^{-1}$. Two sharp peaks, which are linked to domain switching, are observed in the current loop (Fig. 3a). Similar results were obtained on 170 nm- and 500 nm-thick films (10 and 11 μC cm$^{-2}$ in $P_r$, respectively, see Supplementary Figs. 7 and 8). The displacement of a cantilever structure shows a typical butterfly loop (Fig. 3b). At 100 V the vertical displacement at the free end of the cantilever is 700 nm, corresponding to a piezoelectric coefficient $e_{33,f}$ of −5 C m$^{-2}$[45].

Butterfly loops were also observed in electric-field dependence of relative permittivity $\varepsilon_r$ and dielectric losses tan$\delta$ of the 1 μm-thick film (Fig. 3c). Their zero-field values are 450 and 5 %, respectively. These quantities were also measured as functions of the small signal frequency (Fig. 3d). While permittivity slightly decreases with increasing frequency, losses remain practically constant, which are typical signatures of perovskite ferroelectric films[46]. To have a better overview of the results, a table with ferroelectric, piezoelectric and optical properties of the 170 nm, 500 nm and 1 μm-thick films on fused silica substrates is provided in Supplementary Table 1.

### Device application
To demonstrate the suitability of the flash lamp annealing process for piezoelectric applications, we fabricated and characterized a surface haptic device[47]. AF32 glass, which is utilized in the semiconductor and

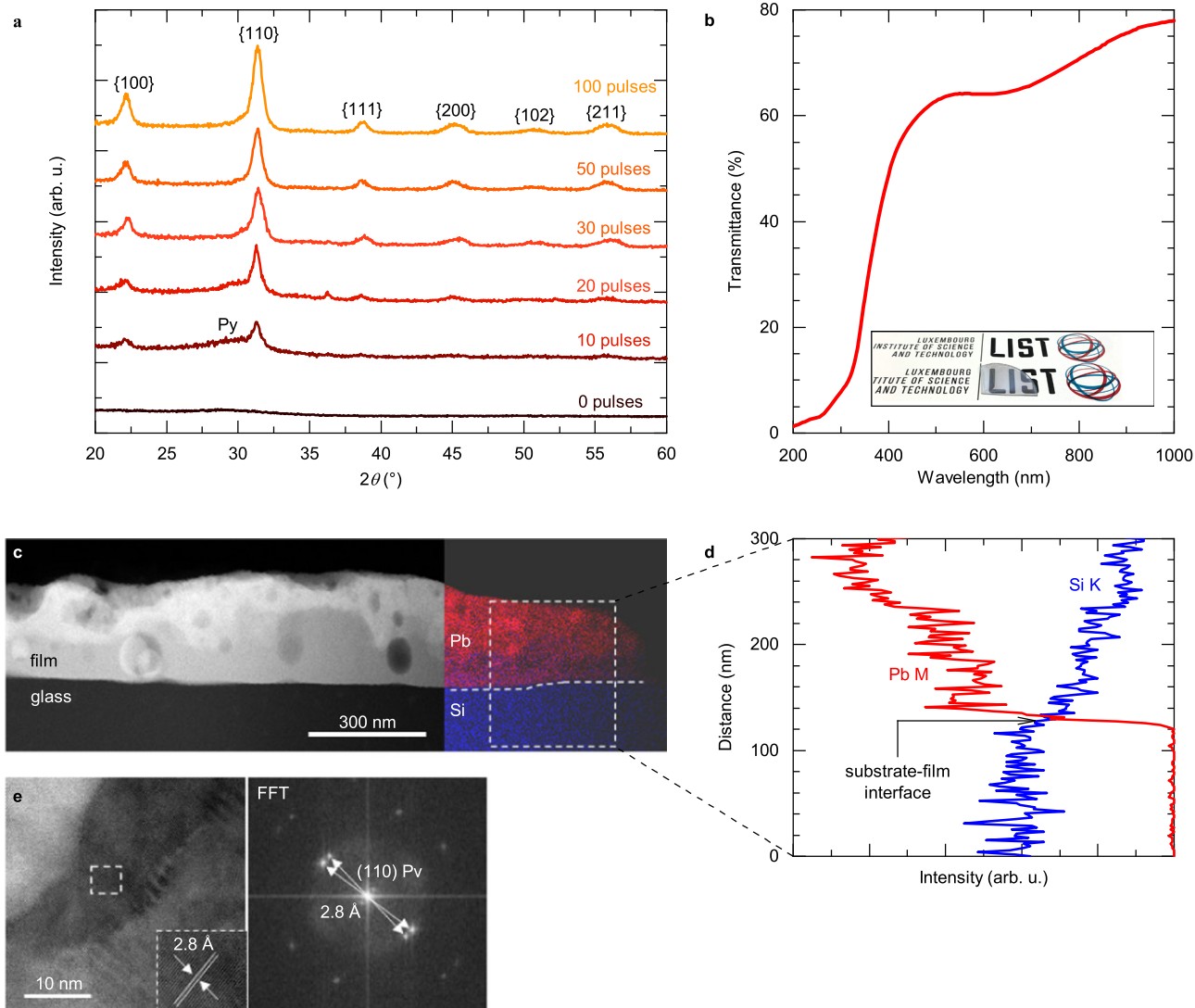

**Fig. 2 | Microstructural and optical characterization of 170 nm-thick PZT films on fused silica glass. a** GIXRD patterns of films annealed with different number of light pulses. In **a** perovskite reflections are denoted according to powder diffraction file (PDF) No 01-070-4264[66]. Py indicates the pyrochlore reflection according to PDF No 04-014-5162[66]. **b** Transmittance of the 50-pulse film. Inset shows its optical appearance. STEM results of the 100-pulse film: **c** cross sectional dark-field STEM image and energy-dispersive X-ray spectroscopy system (EDS) analysis across substrate/film interface (marked by dashed line) using Pb M and Si K line, **d** EDS analysis across substrate-film interface using the Pb M and Si K lines showing partial diffusion of Si into the PZT film. The dashed area in **c** marks the region where the **d** EDS line analysis was performed, **e** high-resolution bright-field STEM image of two perovskite grains with corresponding FFT image showing (110) planes. Flash lamp annealing was performed with energy density, pulse duration and repetition rate of 3 J cm⁻², 130 µs, and 3.5 Hz, respectively.

MEMS industry, was used[48]. A 1 µm-thick film was employed to increase the force of the actuator during its operation (see Supplementary Note 3.2.1). The phase composition and electrical properties of the film are comparable to the films on fused silica, as shown in Supplementary Note 3.

As schematically shown in Fig. 4a, the dimensions of the fabricated haptic device are 15.4 × 3 mm². The thickness of the glass and film are 0.3 mm and 1 µm, respectively. Two actuating areas were created by fabricating 100 nm-thick Pt interdigitated (IDE) electrodes with a gap of 3 µm between the fingers. The distance between these two actuators is 8.4 mm. More details about the haptic device fabrication and measurements are described in the Methods and Supplementary Note 3.2. The out-of-plane displacement at one of the resonance antinodes is shown as a function of frequency in Fig. 4b. At 40.2 kHz the device reveals a peak in deflection, which corresponds to its mechanical resonance at anti-symmetric ($A_O$) Lamb mode. For further info on Lamb mode analysis see Supplementary Note 3.2.3 and Ref. 39.

Its position is at the same value as predicted by finite element modelling, whose details are described in Supplementary Note 3.2.2.

The out-of-plane surface displacement in the *x*-direction (along the length of the device) is shown in Fig. 4c. It was measured with an excitation voltage varying from 20 to 50 $V_{PP}$ at the resonance frequency of 40.2 kHz. Four nodes equally spaced along its length are observed, which is in line with the wave-shape obtained from the modelling (Supplementary Fig. 11). Figure 4d shows a 2D displacement map of the haptic plate excited at 60 $V_{PP}$ (30 $V_{AC}$ + 30 $V_{DC}$) at the resonance. The device exhibits a maximum peak-to-peak deflection of 1.5 µm, beyond the specification of 1 µm, which can be detected by a human finger[37]. It also confirms a neat stationary Lamb wave. $e_{33,f}$ can be extracted by matching the experimental deflection value with modelling[49] and the obtained value is −4.5 C m⁻², which is well in line with the value obtained via cantilever measurements on fused silica (Fig. 3b).

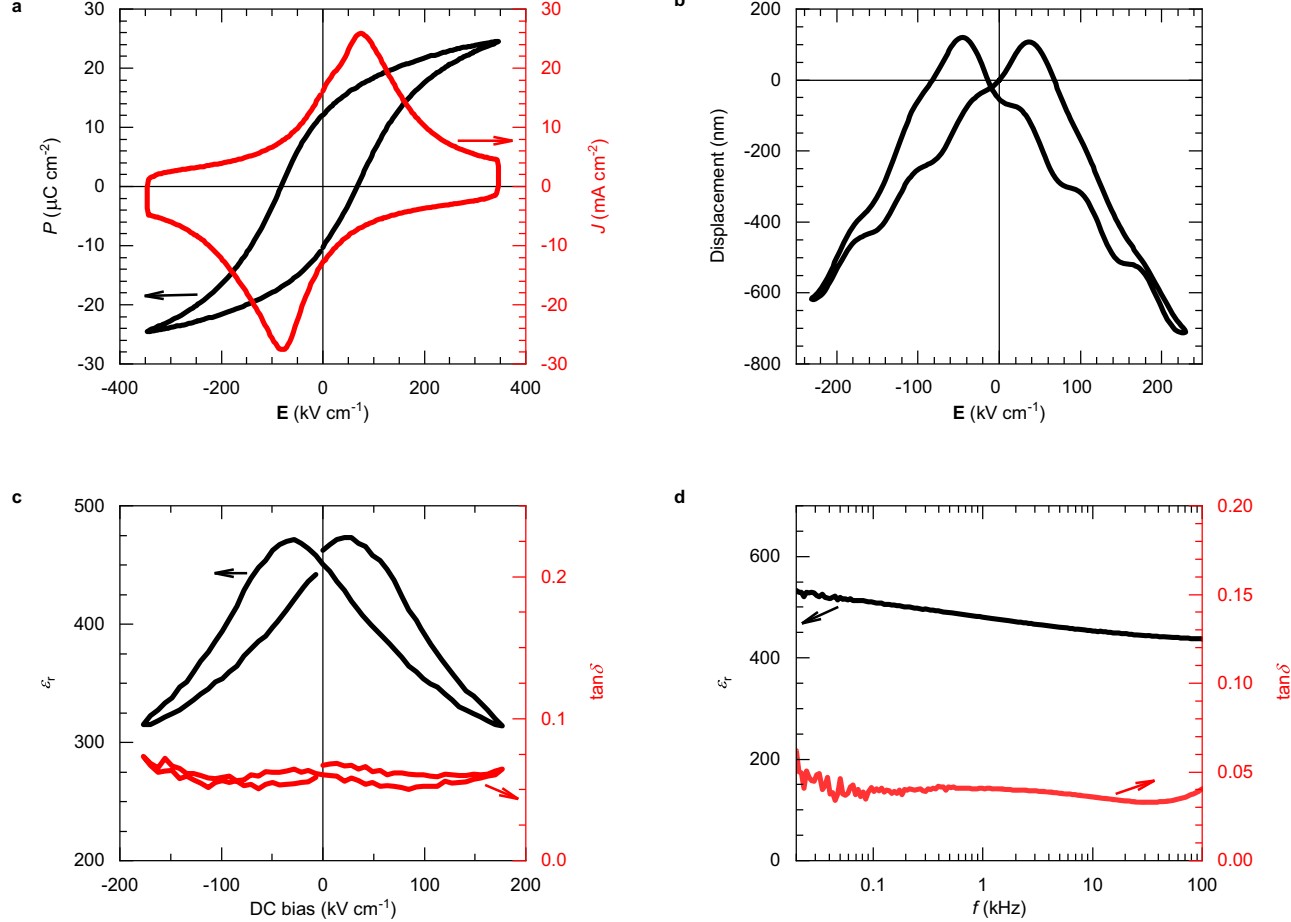

**Fig. 3 | Electromechanical characterization of a 1 μm-thick PZT film on fused silica glass. a** Ferroelectric and **b** displacement characterizations of the PZT film at 100 Hz and 11 Hz, respectively. Dielectric measurements of the PZT film: relative permittivity $\varepsilon_r$ and dielectric losses tan$\delta$ as function of **c** DC bias at 1 kHz and **d** frequency $f$. The samples were processed with 50 pulses per layer with energy density, pulse duration and repetition rate of 3 J cm$^{-2}$, 130 μs, and 3.5 Hz, respectively.

The performance of the device is compared to the previously reported piezoelectric haptic devices on glass substrates based on spin-coated[42] and inkjet-printed[5] PZT films with interdigitated geometry in Supplementary Table 2. The present device consumes 35 mW, which is comparable to the other two devices. Note also that 60 V can be applied to handheld devices by using either a cascade of several application-specific integrated circuits (ASICs) that can increase the output voltage from 3.3 to 100 V or by using an inductor L to make an LC resonator at the resonant frequency[42].

### Flash lamp annealing for films on soda lime glass

To generalize the approach, we also used this flash lamp process to crystallize PZT films on soda-lime glass, which can only sustain temperatures below 400 °C. Because of the lower thermal conductivity of soda lime (1.0 W m$^{-1}$ K$^{-1}$) compared to fused silica (1.4 W m$^{-1}$ K$^{-1}$), the rate of heat transfer is slower, resulting in a higher temperature at the interface between the film and glass. A two-step process, resembling nucleation and growth in CSD process[9], was developed. In the first step, 6 pulses with higher power density of 14.7 kW cm$^{-2}$ (2.5 J cm$^{-2}$ and 170 μs) are applied. In the second step, 240 pulses with a lower power density of 10.0 kW cm$^{-2}$ (2.5 J cm$^{-2}$ and 250 μs) are applied to grow the film at a lower temperature, thereby preventing the occurrence of cracks. For both steps, the repetition rate is set to 0.5 Hz to increase the heat diffusion through the sample. This process is described in Supplementary Note 4, where also more details on phase composition and electrical properties are described (Supplementary Figs. 13 and 14). The

films on soda lime glass also crystallize in the perovskite phase and have $P_r$ of 8 μC cm$^{-2}$, $\varepsilon_r$ of 400, and tan$\delta$ of 2 %, respectively.

## Discussion

Macroscopic ferroelectric and piezoelectric properties of the films crystallized during flash lamp annealing are demonstrated in this work. The previously unreported process of two phases where either nucleation or grain growth of the perovskite phase is dominating has been enabled by the tool design, which allows the creation of high-power pulses (above 20 kW cm$^{-2}$) with high repetition rates (3 Hz)[35,50–52]. A detailed comparison with the literature can be found in Supplementary Note 5.

To better understand the effectiveness of the flash lamp annealing process, we conducted a comparison of the electrical properties between 1 μm-thick flash lamp-processed films and RTA-processed films on fused silica glass[42]. Table 1 displays the corresponding electrical parameters. It is worth noting that, in the RTA process, a hafnia buffer layer was applied to impede Pb diffusion, and that lead titanate oxide (PTO) nucleation was implemented to facilitate growth along {100} and therefore enhance the piezoelectric response[53]. Despite the doubling of the $\varepsilon_r$ value enabled by PTO nucleation, the ferroelectric and piezoelectric properties of the material remain relatively comparable.

It is also interesting to compare flash lamp process with other low-temperature processes of chemical solution-deposited PZT films. While reduced processing temperature has often been achieved,

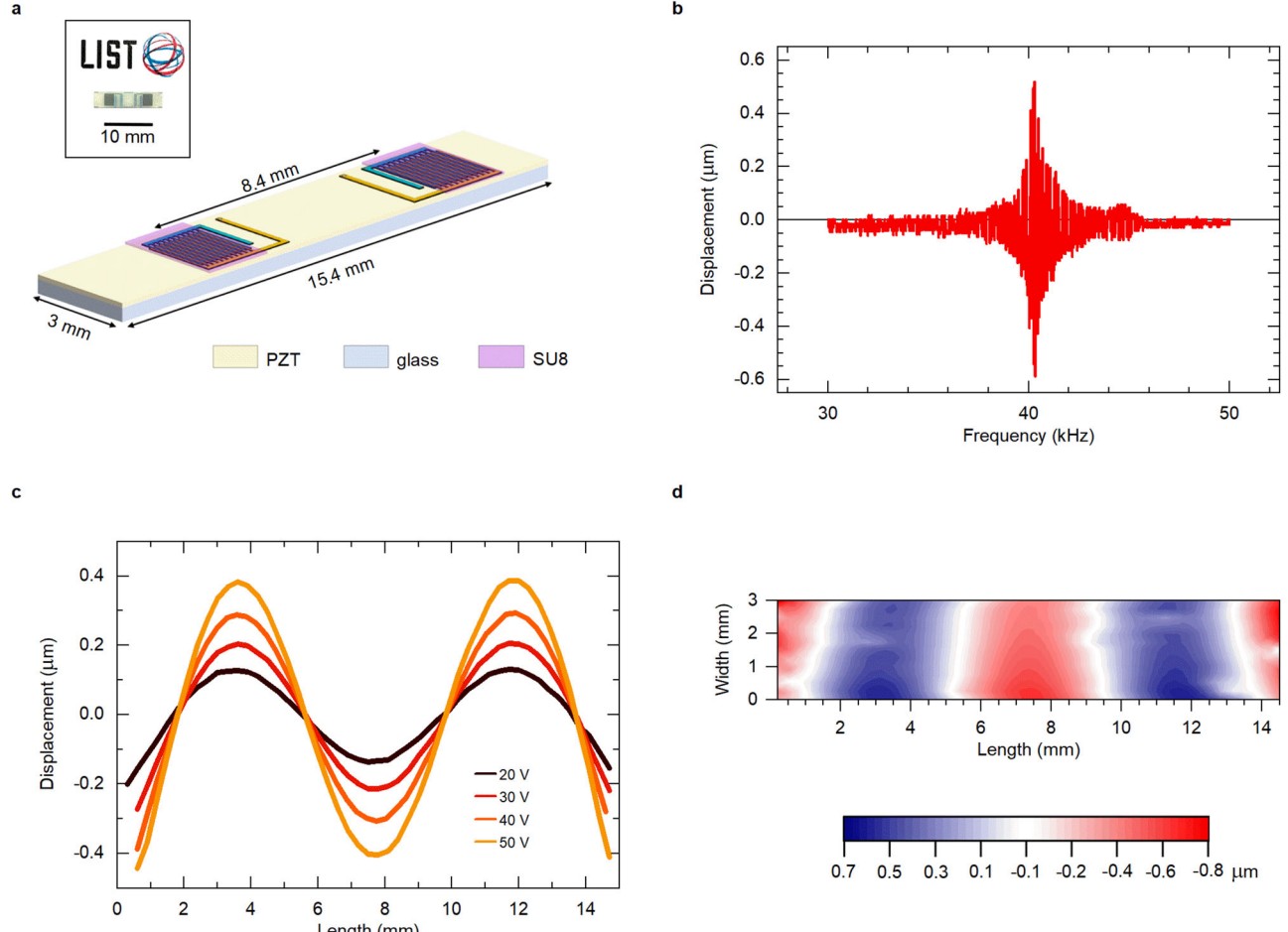

**Fig. 4 | Electromechanical characterization of the haptic device. a** Schematic of the device made of the flash lamp annealed 1 μm-thick PZT on AF32 glass. Two sets of interdigitated Pt electrodes are placed on top to create the actuators. Inset shows its visual appearance. **b** Out-of-plane displacement of the surface measured as a function of frequency at one of the antinodes. 60 $V_{pp}$ (30 $V_{AC}$ + 30 $V_{DC}$) were applied. **c** The displacement along $x$-axis (length of the device) excited at various driving voltages. **d** 2D map of the displacement at 60 $V_{pp}$. **c** and **d** are measured at the resonance frequency of 40.2 kHz.

**Table 1 | Comparison of ferroelectric, dielectric and piezoelectric properties of chemical solution deposited PZT films processed by FLA, RTA and LA**

| Process | Substrate | Structure | $P_r$ (μC cm$^{-2}$) | $\varepsilon_r$ | tanδ | $e_{33,f}$ (C m$^{-2}$) | Ref. |
|---|---|---|---|---|---|---|---|
| RTA | Fused silica | IDE | 16 | 900 | 0.05 | −8 | Glinsek et al.[42] |
| FLA | Fused silica | IDE | 11 | 450 | 0.05 | −5 | This work |
| FLA | AF32 | IDE | 5 | 350 | 0.07 | −5 | This work |
| LA | Platinised silicon | MIM | 28 | – | – | −4 | Fink et al.[20] |

In Ref. 42 a lead titanate oxide nucleation layer was used to promote growth along (100) orientation, which enhances piezoelectric response[53]. MIM stands for metal-insulator-metal structure.

macroscopic piezoelectric properties of such films are seldomly reported, as demonstrated in a recent review paper[12]. The highest value reported so far in low-temperature films is $d_{33,f}$ = 49 pm V$^{-1}$ in laser-annealed (LA) PZT films on platinised silicon[20]. For comparison, this $d_{33,f}$ value was translated into $e_{33,f}$ (−3.9 C m$^{-2}$) by taking the typical value of Young's modulus (80 GPa) of PZT[54,55], which is slightly lower than those of flash lamp annealed films.

Given that the electrical properties are comparable, it is crucial to assess the processes. RTA necessitates a buffer and a processing time of tens of minutes and is incompatible with low-temperature glass. Laser annealing, on the other hand, requires non-transparent bottom electrodes (such as Pt or LaNiO$_3$) for nucleation, without which films cannot crystallize[19,20]. The small laser spot size also requires raster

scanning of the laser beam, leading to inhomogeneities in the films. These limitations hinder process efficiency, resulting in reduced productivity. In contrast, the flash lamp process offers direct and rapid growth in an ambient environment without relying on any buffer or nucleation, while also being highly compatible with roll-to-roll production thanks to its large irradiation area. These advantages make it a high-output process.

The previous comparison and the results presented above provide evidence that the flash lamp process facilitates the direct growth of piezoelectric films on a variety of glass substrates, which exhibit electromechanical properties meeting the requirements for piezoelectric applications, as demonstrated by the successful realisation of the haptic rendering device. A distinctive

advantage of the flash lamp process is its high compatibility with digital inkjet printing and large-scale roll-to-roll manufacturing, which puts it ahead of other low-temperature processes. Through the interaction of light with material, the process allows for the growth of perovskite films without nucleation layers. Compared to aerosol-based deposition techniques[56–59], the FLA-based process enables in-situ crystallization of the perovskite phase and eliminates the need for post-annealing step to improve the properties.

As a conclusion, we proved in this paper that a well-defined flash lamp annealing process enables the crystallization of piezoelectric perovskite thin films deposited on glass substrates that cannot withstand temperatures larger than 400 °C. All these points are strong assets in favour of perovskite piezoelectric films for future transparent and flexible electronics.

## Methods

### Processing and deposition of PZT solution

$Pb(Zr_{0.53}Ti_{0.47})O_3$ solution was prepared via the standard 2-methoxyethanol (99.8 %, Sigma-Aldrich) route using freeze-dried lead(II) acetate trihydrate (99.99 %, Sigma-Aldrich), titanium(IV) iso-propoxide (97%, Sigma-Aldrich) and zirconium(IV) propoxide (70 % in propanol, Sigma-Aldrich) as starting compounds[60]. Concentration ($C_{Zr} + C_{Ti}$) was 0.3 M with a stoichiometry of metal cations Zr:Ti = 0.53:0.47, which corresponds to the morphotropic phase boundary composition with 10 mol.% Pb excess. The total volume of the solution was 50 mL. For more details see Ref. 60. The solutions were spin-coated on glass substrates for 30 s at 3000 rpm with an acceleration of 500 rpm s$^{-1}$. Three types of glasses were used: fused silica (2″ and 0.5 mm-thick, SIEGERT Wafer, fused silica)[61], AF32 (2″ and 0.3 mm-thick, SCHOTT, AF32)[48] and soda lime (38 × 26 × 1 mm$^3$, Marienfield, ref 1100020 (microscope slides) and 38 × 25 × 1 mm$^3$, Epredia, ref AB00000102E01MNZ10 (microscope slides))[62]. Drying and pyrolysis were performed on hot plates for 2 min each at 130 °C and 350 °C, respectively. The deposition-drying-pyrolysis cycle was repeated four times to obtain 170 nm-thick amorphous PZT films.

### Flash lamp annealing

Crystallization of the amorphous films was performed with a flash lamp annealer (Pulseforge Invent, Novacentrix). The parameters to control the annealing process include energy density, pulse duration, and repetition rate. The respective values for a standard one-step process (fused silica and AF32 glasses) were 3 J cm$^{-2}$, 130 μs and 3.5 Hz. Note that the energy density was experimentally measured using a bolometer, and variation within 5 % in lamp energy density is observed in a fast multipulse process due to the lamp stability. In a two-step process (soda lime glass), films were annealed for a few pulses at high power density followed by numerous pulses at lower power density. This two-step process is described in more detail in Supplementary Note 4.1.

### Microstructural and optical characterization

Phase composition and orientation of the films were investigated with GIXRD and standard $\theta$-$2\theta$ XRD, which were performed on a D8 Discover diffractometer (Bruker) using Cu-K$_\alpha$ radiation. The incidence angle of GIXRD was 0.5°. Both GIXRD and standard $\theta$-$2\theta$ XRD patterns were recorded in the $2\theta$-range from 20° to 60° with a step-size of 0.02°.

Samples for STEM analyses were prepared by grinding, dimpling and final Ar milling (Gatan PIPS Model 691, New York, NY, USA). STEM studies were carried out using a probe Cs-corrected Jeol ARM200 CF (Jeol, Tokyo, Japan) equipped with a Centurio energy-dispersive X-ray spectroscopy system (Jeol, Tokyo, Japan), operated at 200 kV.

Optical spectra were obtained using a UV/Vis spectrophotometer (Spectro L1050, PerkinElmer).

### Electromechanical characterization

Interdigitated Pt electrodes were deposited and patterned on top of the films with lift-off photolithography using direct laser writing (MLA 150, Heidelberg Instruments), as described in our previous work[42]. Pt was chosen as it is the most mature electrode material for piezoelectric thin films. Two sizes of IDE electrodes were used. Finger width, length of digits facing each other, interdigital gap, and pairs of digits were 5 μm (5 μm), 1730 μm (370 μm), 3 μm (3 μm), and 615 (50), respectively for large (small) IDEs 100 nm-thick Pt electrodes were sputtered with a MED 020 metallizer (BalTec).

Electrical characterization was performed using a TF Analyzer 2000 (aixACCT). The polarization was measured as a function of electric field $P(\mathbf{E})$ with a triangular waveform at 100 or 10 Hz. Bipolar fatigue cycling was performed at 1 kHz, with a triangular fatigue voltage of 100 V.

The converse piezoelectric response of 500 nm PZT film on fused silica glass (Fig. 3b) was measured with a thin-film sample holder unit (aixACCT) and an interferometer (SP120/2000, SIOS)[63]. Cantilevers with dimensions of 25 × 3.4 mm$^2$ were cut from the wafers using a wire saw. The geometry of the IDE capacitor was the same as for large IDE. The converse piezoelectric coefficient $e_{33,f}$ was extracted as described in Ref. 45. A Young's modulus of 73 GPa has been used for the fused silica substrate.

Relative permittivity $\varepsilon_r$ and dielectric losses tan$\delta$ of 1 μm-thick film were measured as functions of DC voltage with a probing AC signal of 0.5-1 V at 1 kHz. The IDE used for this measurement (Fig. 3c) has finger width of 10 μm, a length of digits facing each other of 872 μm, an interdigital gap of 10 μm, and 502 pairs of digits, respectively, corresponding to an effective area of 0.86 mm$^2$. Capacitance $C$ and tan$\delta$ were measured as functions of frequency $f$ with an Agilent E4990A impedance analyzer with an AC voltage of 1 V. Corresponding $\varepsilon_r$ was calculated from capacitance. The IDE used for this measurement (Fig. 3d) has the parameters of 10 μm, 2340 μm, 10 μm and 51, respectively, corresponding to an effective area of 0.209 mm$^2$.

### Haptic device

The 1 μm-thick PZT film was prepared on AF32 glass, as described above. Based upon FEM, the IDE Pt electrodes had an overall dimension of 2050 × 2340 μm$^2$ and were distributed on the 15.4 × 3 mm$^2$ device as schematically shown in Fig. 4a. Copper wires bonded using silver epoxy were used to connect voltage and ground electrode pads. The haptic device was placed on suspended flexible foam tape and was connected to a waveform generator (33210 A, Keysight) via an amplifier (WMA-300, Falco Systems). The out-of-plane displacement was recorded using a laser Doppler vibrometer (OFV-5000, Polytec). A computer-controlled $x$-$y$ stage was used to move the haptic device for line scans and 2D mapping. The whole set-up was controlled with the vibrometer via a LabVIEW program, which has been also demonstrated in a previous paper[42]. Note that the two actuators were connected in parallel, therefore, the voltage applied to each actuator is the same. Note also that all the measurements in Fig. 4 were done with a unipolar voltage.

## Data availability

The data that supports the findings of the study are included in the main text and supplementary information files. The raw data generated in this study have been deposited in the Zenodo repository[64] under DOI 10.5281/zenodo.10622727.

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

## Acknowledgements

Luxembourg National Research Fund (FNR) is acknowledged for the financial support through project FLASHPOX (C21/MS/16215707, grant recipient: S.Gli.). L.S and A.B.M. acknowledge financial support from the FNR under the project PACE (PRIDE/17/12246511/PACE). A.B. and B.K. acknowledge funding by the Slovenian Research Agency (projects J2-3041, J2-2497, grant recipient: A.B.). Barnik Mandal and Poorani Gnanasambandan are acknowledged for performing SEM. Some of the data from this manuscript was previously published in the thesis of L.S.[65].

## Author contributions

L.S. and J.C. contributed equally to the acquisition, analysis, interpretation of data and writing of the work. A.B. contributed to the acquisition and analysis of data. A.B.M., B.K. and S.G. contributed to the acquisition of data. E.D. contributed to the conception of the work. S.Gli. contributed to the conception of the work and substantively revised it.

## Competing interests

The authors declare no competing interests.
