## [Peer Review File · Nature Communications]

Crystallization of piezoceramic films on glass via flash lamp annealingEditorial Note: Parts of this Peer Review File have been redacted as indicated to remove third-party material where no permission to publish could be obtained.

REVIEWER COMMENTS

Reviewer #1 (Remarks to the Author):

1. Other additive manufacturing techniques are also capable of coating various piezoceramics on glass, such as aerosol deposition and other spraying techniques. These techniques are also high-speed and scalable. In recent, many of them have been industrialized. Most importantly, despite these techniques requiring no annealing process, they have shown decent or great performances in piezoelectric properties. What are the novelty and advantages of the FLA method compared to those techniques?
2. Detailed explanation of crystalline planes revealed in XRD results is needed to better characterize the perovskite crystalline structure. What about the XRD result of 500 nm-thick film?
3. The author mentioned the average transmittance is 95%. How did you calculate the average value? What was the standard wavelength when averaging the transmittance?
4. It's hard to find figures showing the partial diffusion of silicon into the PZT film.
5. The authors mentioned that "More details about the haptic device ... are described in Supplementary Section4". The description is in Section 3.
6. Real photos of the haptic device are needed.
7. Ferroelectric and piezoelectric properties of 170 nm-, 500 nm-, and 1 um-thick film cases were examined. These cases had different wake-up cycles of 1.1×10^6 , 10^3 , and 10^4 cycles, respectively. Why did they have different number of cycles? And, what were the maximum polarization values of 170 nm and 1 um cases? What were the transmittance of 500 nm and 1 um cases?
8. Why did you use 1 um-thick film case when fabricating the haptic device, not using 170 nm and 500 nm cases?
9. For the transmittance, the authors used and explained the 170 nm-thick film case. For the ferroelectric and piezoelectric properties, they mainly presented the 500 nm-thick film case. For the haptic device, they used the 1 um-thick film case. Why did you use and examine different cases with no consistency?

Reviewer #2 (Remarks to the Author):

The authors have made substantial progress in developing a low-temperature flash lamp thermal treatment process of a few seconds for growth of piezoelectric lead zirconate titanate films on glass. They declared good piezoelectric properties obtained on glass substrate, showing potential in realizing applications of transparent piezoelectric devices.

The following suggestions could help the authors to improve the manuscript.

1. It is not clear how the authors obtained the experimental of 2D map of the displacement as shown in Fig. 4, which is an important experimental outcome as declared. The large displacement of 1.5 um is

- obtained at 60 Vpp. Is the voltage too high? How about comparison with the state of the art results in the literature?
2. The large displacement has been obtained at resonance condition. Which resonance mode? If it is related to Lamb wave, how about the dispersion curves?
 3. The title, "Low-temperature growth of piezoceramic films on glass" is too broad for this paper, which should be more specific. Relevant key words, such as flash lamp, and/or the material, PZT, should be added in the title.
 4. Why the authors select Pt instead of a transparent electrode on the top, although interdigital electrode configuration is used?
 5. In Page 6, Lines 117-118, the "Further details on processing are given in Supplementary Section 2", should be changed into "Section 4.1 Processing and deposition of PZT solution", as the details on processing are not mentioned in the Supplementary Section 2.
 6. In Pages 8-9, the content for comparing various annealing processes including the comparison table could be moved from the section titled, "Electromechanical Characterization" to the discussion section.
 7. In Page 10, Lines 208, 217 and 228 mentioned Supplementary Section 4. However, "Section 2.4 Device Application" is about the film on AF32 glass, which correspond to the Supplementary Section 3. Note Page 11, Line 237, where the Supplementary Section 5 should be Supplementary Section 4. The authors are reminded of checking and correcting citations for the corresponding supplementary sections.
 8. In Page 12, there mentioned two kinds of FLA process which include one-step (Line 272) and two-step (Line 257) process. The one-step process is introduced in details, but not for the two-step process, and it is not clear why the two-step process was used. Any advantages compared with one-step process?
 9. Figures 2 and 3 both show the PZT film on fused silica glass. However, Figure 2 shows the microstructure and optical characterization for the film of 170-nm in thickness, and Figure 3 provides electromechanical characterization results for 500-nm in thickness. Why do not provide the results for the same thicknesses?
 10. In Supplementary Section 2, Page 8, Line 90, it mentioned "50 pulses exhibit the optimal electrical property". In the main text, Page 6 Lines 125-126, "100 pulses indicating the enhanced crystallinity", but not mentioned which pulses has optimal property. Should the point in the supplementary info be mentioned in the main text for clarity.

Reviewer #3 (Remarks to the Author):

The authors investigate the so-called low-temperature flash lamp process for the growth of piezoelectric lead zirconate titanate films (PZT), in which various types of glasses are used as substrates. The main limitation in processing PZT films on glass is applying temperatures higher than 700 °C for the effective annealing of the films while keeping the glass substrate at temperatures below 600 °C, which has been successfully overcome in this study. While the overall study is well designed, the data are meticulously analyzed, and the interpretation and methodology are sound, I have reservations about the novelty of the findings and outlook. To my opinion they do not meet the standards of novelty and significance expected for publication in Nature Comm.

Significance: The central question addressed by the study is important, and the use of FLA is beneficial for high-throughput, and large-scale roll-to-roll production. However, the reported findings do not present novel insights or unexpected outcomes. The manuscript lacks the transformative impact that is expected from publications in high-impact journals.

Novelty and Specific Concerns: The research presented in this manuscript appears to be a continuation of previous studies related to the FLA of PZT films and does not introduce new concepts or approaches.

The unique aspects of the authors' work that differentiate it from the existing literature are poorly highlighted. As briefly mentioned in the manuscript, the presented concept is already available in the literature:

- DOI: 10.1143/JJAP.41.2630

- DOI: 10.1016/j.jeurceramsoc.2020.07.052

- DOI: 10.1002/adma.202303553

- DOI: 10.1111/jace.14272

- Thesis: Ouyang, Jing, "Enhanced Piezoelectric Performance of Printed PZT Films on Low Temperature Substrates" (2017)

- Thesis: Marotta, Amanda R., "Printable Thin-Film Sol-Gel Lead Zirconate Titanate (PZT) Deposition Using NanoJet and Inkjet Printing Methods" (2019).

There have already been reports on PZT films spin coated and printed from the ink-based solution, annealed with FLA on glass and plastic substrates. There is no clear evidence if the ink formulation, FLA processing parameters, or deposition methods in the manuscript are conceptually different compared with the reported results.

The conceptualization of the study as a "low-temperature growth" process is considered wrong, since the film is deposited and annealed at high temperatures, according to the simulations, while only the substrate remains at low temperatures.

The variation of the film thickness from 170 nm to 500 nm and 1 μm for different types of substrates is confusing.

Recommendation:

Based on the concerns outlined above, I suggest that the authors submit their manuscript to a more specialized journal that focuses on ceramic film processing. This would provide a more appropriate platform for the dissemination of their work to a targeted audience who would better appreciate the incremental advancements made by this study.

Crystallization of piezoceramic films on glass via flash lamp annealing

Submission ID: NCOMMS-23-20861

We thank the reviewers for providing us with valuable comments and constructive criticism. We considered all of them and implemented changes in the manuscript accordingly (highlighted in yellow). The comments are addressed in details hereafter, point by point.

Reviewer 1

Comment 1.1

Other additive manufacturing techniques are also capable of coating various piezoceramics on glass, such as aerosol deposition and other spraying techniques. These techniques are also high-speed and scalable. In recent, many of them have been industrialized. Most importantly, despite these techniques requiring no annealing process, they have shown decent or great performances in piezoelectric properties. What are the novelty and advantages of the FLA method compared to those techniques?

Response 1.1

We thank the reviewer for this comment. Indeed, other additive manufacturing techniques allow for deposition of piezoceramics on glass. As explained below, these techniques are intrinsically different in processing and in the values of piezoelectric coefficients they can achieve.

In-situ crystallization (novelty). Aerosol-based deposition relies on powders to form thick films ranging from several to tens of μm . The quality of these films depends heavily on the properties of the powders used, which are fabricated at high temperatures (beyond $1000\text{ }^{\circ}\text{C}$)¹, inducing extra costs and process steps. Spraying techniques allows for the deposition of both thick ($> 5\text{ }\mu\text{m}$) and thin films ($< 3\text{ }\mu\text{m}$) through the spraying of powders or solutions, respectively. In the latter case, high-temperature annealing is needed to crystallize the films. Our work, on the other hand, enables in-situ crystallization of oxide films while keeping the substrate at low temperature. This distinguishes it from aerosol deposition and spraying processes.

Good piezoelectric properties (advantage). Non-annealed PZT layers deposited via aerosol-based techniques show low piezoelectric and ferroelectric response due to structural imperfections and small crystallite size². A post-annealing above 600 °C is necessary to improve the properties^{2,3}, see recent review articles on this topic^{2,4}. As an example we take the work of Sadl et al.⁵, who prepared 0.65Pb(Mg_{1/3}Nb_{2/3})O₃ – 0.35PbTiO₃ (PMN-PT) on stainless steel substrates via aerosol deposition. The as-deposited films did not show piezoelectric response, while the films post-annealed at 500 °C had $d_{33,f}$ of 25 pm V⁻¹ ($e_{33,f}$ of ~2 C m⁻² assuming Young’s modulus for PZT of 80 GPa). As-deposited PZT films in our work show $e_{33,f}$ of 5 C m⁻².

To clarify this in the main manuscript we added a sentence in the last part of the discussion: “Compared to aerosol-based deposition technique⁴⁸⁻⁵¹, the FLA-based process enables in-situ crystallization of perovskite phase and eliminates the need for post-annealing step to improve the properties.” Note that ref.^{1-3,5} were also added.

Comment 1.2

Detailed explanation of crystalline planes revealed in XRD results is needed to better characterize the perovskite crystalline structure. What about the XRD result of 500 nm-thick film?

Response 1.2

The text on page 6 of the main manuscript, has been changed to “After 10 pulses, {100} and {110} reflections of the perovskite phase appear in the grazing incidence X-ray diffraction pattern (Fig. 2a) of 170 nm-thick films. Reflection of the non-piezoelectric pyrochlore phase (a small peak at $2\theta = 29^\circ$), which is kinetically stabilized during annealing³², disappears when the number of pulses is increased to 30.”

Additionally, a PDF No 04-014-5162⁶ reference for the pyrochlore phase has been added to the captions of Figure 2 and all relevant Figures in the Supplementary Information.

The XRD pattern of the 500 nm-thick film on fused silica glass is now included in Figure S2 in the Supplementary Information and is shown below. The pattern is in line with the 1 μm film, with lower intensities due to lower thickness.

Figure 1: **XRD study.** θ - 2θ XRD patterns of flash lamp annealed 500 nm and 1 μm PZT films on fused silica glass. The films were processed with 50 pulses per layer (3 and 6 crystallizations in total, respectively). Parameters of each pulse are reported in caption of Fig. 2 in the main manuscript. The powder diffraction files (PDF) No 01-070-4264 and 04-014-5162⁶ have been used to identify the perovskite and pyrochlore phases, respectively.

Comment 1.3

The author mentioned the average transmittance is 65%. How did you calculate the average value? What was the standard wavelength when averaging the transmittance?

Response 1.3

The average transmittance mentioned in the article was obtained by taking the average of transmittance values between 500 and 700 nm, excluding the absorption edge from the calculation.

Instead of indicating the above average value in the manuscript, we took the opportunity to correct it by indicating the transmittance at a wavelength of 550 nm, which is the standard wavelength used to characterize the optical transparency of transparent materials (close to a peak sensitivity of a human eye)⁷. The transmittance at 550 nm is 64%.

This correction has been done on page 7 of the main manuscript: “Transmittance of the 170 nm-thick films (50 pulses) at a wavelength of 550 nm is 64%, as shown in Fig. 2b, together with its visual appearance.”

Comment 1.4

It's hard to find figures showing the partial diffusion of silicon into the PZT film.

Response 1.4

In the Figure below we show a detailed map of EDS analysis of the film cross-section (already shown in Figure 2c in the main article). In the marked area we performed the analysis where we show that the Si signal extends into the PZT layer.

Figure 2: **TEM and EDS analysis of 170 nm PZT thin film on fused silica glass.** EDS analysis across substrate-film interface using the Pb M and Si K lines showing partial diffusion of Si into the PZT film. The dashed area in a) marks the region where the b) EDS line analysis was performed.

The Figure was added to the Supplementary as Figure S8 and we modified the sentence in the main manuscript to refer to it (Section 2.2): “It is also observed that there was partial diffusion of silicon into the PZT film, at a depth of approximately 100 nm (see Supplementary Fig. S8 for more details).”

Comment 1.5

The authors mentioned that “More details about the haptic device ... are described in Supplementary Section4”. The description is in Section 3.

Response 1.5

We apologize for this mistake. Section numbering was corrected.

Comment 1.6

Real photos of the haptic device are needed.

Response 1.6

A photo of the haptic device has been added as an inset in Figure 4a as is shown below.

Figure 3: **Electromechanical characterization of the haptic device.** a) Schematic of the device made of the flash lamp annealed 1 μm -thick PZT on AF32 glass. Two sets of interdigitated Pt electrodes are placed on top to create the actuators. Inset shows its visual appearance. b) Out-of-plane displacement of the surface measured as a function of frequency at one of the antinodes. 60 V_{pp} (30 V_{AC} +30 V_{DC}) were applied. c) The displacement along x -axis (length of the device) excited at various driving voltages. d) 2D map of the displacement at 60 V_{pp} . c) and d) are measured at the resonance frequency of 40.2 kHz.

Comment 1.7

Ferroelectric and piezoelectric properties of 170 nm-, 500 nm-, and 1 μm -thick film cases were examined. These cases had different wake-up cycles of 1.1×10^6 , 10^3 , and 10^4 cycles, respectively. Why did they have different number of cycles? And, what were the maximum

polarization values of 170 nm and 1 μm cases? What were the transmittance of 500 nm and 1 μm cases?

Response 1.7

Different number of cycles. Wake-up cycles were performed to open polarization loops and reach the maximum remanent polarization (P_r) value. We think that the wake-up effect is due to charged defects present in the films, which pin polarization at low electric fields and get re-distributed upon electric-field cycling. Different number of wake-up cycles in different films could be due to different concentration of defects in these films and/or their different pinning energy. However, we think that further studies of this effect are beyond the scope of this article.

To make this point clearer, we added the following sentence to Section 2.3 of the main manuscript: “Different number of wake-up cycles in different films could be due to different concentration of defects in the films and/or their different pinning energy.”

Maximum polarization values. The maximum polarization values of 170 nm-thick and 1 μm -thick films are 19 and 24 $\mu\text{C cm}^{-2}$, respectively, and are presented in Table 1. We added this table as Table S1 in the Supplementary.

Transmittance. The 550 nm-transmittance of 500 nm and 1 μm films is 56% and 49%, respectively, see Figure below. This Figure was added as Figure S5 in Supplementary Information. Note that the former Figure S4 was removed as it became redundant with the inset of the new Figure.

Figure 4: **Transmittance of 170 nm, 500 nm and 1 μm -thick PZT thin films on fused silica glass.** Inset shows the optical appearance of the 1 μm -thick film. The films were processed with 50 pulses per layer (3 and 6 crystallizations in total, respectively). Parameters of each pulse are reported in caption of Fig. 2 in the main manuscript.

Comment 1.8

Why did you use 1 um-thick film case when fabricating the haptic device, not using 170 nm and 500 nm cases?

Response 1.8

In general, piezoelectric films show increased electromechanical response with increasing thickness due to larger contributions of domains⁸. In the case of piezoelectric actuators with interdigitated (IDE) geometry, additional benefit of using thicker film is an increased in-plane force F_3 exerted by the piezoelectric layer upon applied electric field E_3 . The force is expressed as:

$$F_3 = -e_{33,f}E_3A, \quad (1)$$

where E_3 is an in-plane electric field, $e_{33,f}$ is an effective piezoelectric coefficient and A is a cross-section. In IDE geometry E_3 roughly equals to an applied voltage U divided by a gap a between the fingers, while A equals to a film thickness t_f multiplied by a finger length l . The above equation can be therefore re-written as:

$$F_3 = -e_{33,f} \frac{U}{a} lt_f, \quad (2)$$

from which it follows that the force exerted by IDE piezoelectric actuator (at constant voltage) can be increased by increasing film's thickness (and decreasing the gap between the fingers). Considering these points and ease of processing, we defined 1 μm -thick PZT film as a good compromise.

To make this point clear, we added the above discussion in Supplementary Information in Section 3.2.1 "Thickness of piezoelectric film". We added the following sentence in the main manuscript, first paragraph in Section 2.4: "A 1 μm -thick film was employed to increase the force of the actuator during its operation (see Supplementary Section 3.2.1).".

Comment 1.9

For the transmittance, the authors used and explained the 170 nm-thick film case. For the ferroelectric and piezoelectric properties, they mainly presented the 500 nm-thick film case. For the haptic device, they used the 1 um-thick film case. Why did you use and examine different cases with no consistency?

Response 1.9

We thank the reviewer for pointing out this inconsistency. The reason for different thickness is the fact that thinner films were made during process development. For electromechanical and device characterization we focused on thicker films as they have higher functional response. To improve clarity of the manuscript, we made the following changes:

- 1) For process development part (Figure 2 and soda lime glass) we focus on 170 nm-thick films (as in the initial submission).
- 2) For electromechanical characterization (Figure 3) and device characterization (Figure 4) we focus on 1 μm films.
- 3) To have better overview of the results, we added a table with ferroelectric, piezoelectric and optical properties of the 170 nm, 500 nm and 1 μm -thick films on fused silica substrates. This information is in Table S1 of an added Section 2.3 of the Supplementary Information and is also shown below.

Table 1: **Properties of 170 nm, 500 nm and 1 μm -thick PZT films on fused silica glass.** Remanent and maximum polarization (P_r and P_{max}) at an applied voltage of 150 V, relative permittivity, and dielectric losses (ϵ_r and $\tan\delta$), piezoelectric coefficient $e_{33,f}$, and transmittance (T) at a wavelength of 550 nm. The films were processed with 50 pulses per layer with energy density, pulse duration and repetition rate of 3 J cm⁻², 130 μs , and 3.5 Hz, respectively. Large IDEs were used in these measurements, corresponding to an effective area of 0.36 mm².

Film thickness	P_r ($\mu\text{C cm}^{-2}$)	P_{max} ($\mu\text{C cm}^{-2}$)	ϵ_r	$\tan\delta$	$e_{33,f}$ (C m ⁻²)	T (%)
170 nm	10	19	200	0.05	-2	64
500 nm	11	21	270	0.05	-5	56
1 μm	12	24	450	0.05	-5	49

Note that some of the P_r (from 11 $\mu\text{C cm}^{-2}$ to 12 $\mu\text{C cm}^{-2}$) and E_c (from 95 kV cm⁻¹ to 68 kV cm⁻¹) values in the text manuscript had to be changed. Also note that Figure 3a and b of the original submission has been moved to Figure S10 of the Supplementary Information and that original Figure S8 has been removed to avoid redundancy arising from the changes.

Reviewer 2

The authors have made substantial progress in developing a low-temperature flash lamp thermal treatment process of a few seconds for growth of piezoelectric lead zirconate titanate films on glass. They declared good piezoelectric properties obtained on glass substrate, showing potential in realizing applications of transparent piezoelectric devices.

Response:

We thank the reviewer for their positive feedback.

Comment 2.1

It is not clear how the authors obtained the experimental of 2D map of the displacement as shown in Fig. 4, which is an important experimental outcome as declared. The large displacement of 1.5 μm is obtained at 60 V_{pp}. Is the voltage too high? How about comparison with the state-of-the-art results in the literature?

Response 2.1

2D map. As pointed out in the original Supplementary Information the out-of-plane displacement was recorded using an OFV-5000 Polytec laser doppler vibrometer in velocity mode. A computer-controlled (LabVIEW) x-y stage was used to incrementally move the haptic device and record the displacement values along the length and width to obtain either line scans or 2D maps. Note that photos of the set-up and its detailed description were already presented in our previous publication⁹, which has already been referenced in the original manuscript.

To make this clearer, we moved a description of the device characterization into the Methods part of the main manuscript (now “4.5 Haptic device”).

Applied voltage and comparison with literature. As the reviewer correctly noted, 1.5 μm displacement was obtained with 60 V_{pp}. This voltage is in fact not too high and can be applied in real applications like mobile phones as long as the voltage remains below 100 V. As pointed out in our previous publication⁹, one possibility is to cascade several application-specific integrated circuits (ASICs) that can increase the output voltage from 3.3 to 100 V (such as MM3097 ASICs from Mitsumi). The other solution is to implement an inductor L to make an LC resonator at the resonant frequency.

The closest previous works are the two reports by our group on spin-coated⁹ and inkjet-printed haptic devices¹⁰ (both conventionally annealed). The three devices are similar in geometry (~3 mm x 15 mm) and have interdigitated electrode structure, which makes the comparison of device performance straightforward. Several points stem from Table 2: 1) The device demonstrated in this work operates at lower frequency. This is mainly due to thinner substrate (300 μm vs. 500 μm). 2) Device in this work needs lower U_{rms} to achieve 1 μm displacement. This is mainly due to decreased gap between the fingers (3 μm vs. 10 μm , see Equation (2)) and lower substrate thickness. 3) Higher total capacitance in the current device is mainly due to a combination of smaller gap and finger width (5 μm vs. 10 μm). Most importantly, this device shows similar power consumption (35 mW) compared to the other two devices, which confirms its high quality.

Table 2: **Comparison of piezoelectric thin-film haptic devices on glass with interdigitated geometry.** f – resonant (operating) frequency; U_{rms} – root mean square (rms) voltage at 1 μm deflection; C_{device} – capacitance of the device; P_{cons} – power consumption estimated as $P_{\text{cons}} = C f (U_{\text{rms}})^2$. In all three cases thickness of PZT was 1 μm .

Device	Glass	Glass thickness (μm)	f (kHz)	U_{rms} (V)	C_{device} (pF)	P_{cons} (mW)
Spin-coated ⁹	fused silica	500	73.0	43	240	32
Inkjet-printed ¹¹	fused silica	500	63.3	46	230	31
This work	AF32	300	40.2	34	760	35

We added the above Table and a corresponding text to a Supplementary Section 3.2.3 “Device performance”. In the main manuscript we added the following text in the last part of Section 2.4 “Device application”: “Performance of the device is compared to the previously reported piezoelectric haptic devices on glass substrates based on spin-coated³³ and inkjet-printed⁵ PZT films with interdigitated geometry in Table S2 in Supplementary. Present device consumes 35 mW, which is comparable to the other two devices. Note also that 60 V can be applied to handheld devices by using either a cascade of several application-specific integrated circuits (ASICs) that can increase the output voltage from 3.3 to 100 V or by using an inductor L to make an LC resonator at the resonant frequency³³.”

Comment 2.2

The large displacement has been obtained at resonance condition. Which resonance mode? If it is related to Lamb wave, how about the dispersion curves?

Response 2.2

As correctly pointed out by the reviewer, our device is operating in a resonance mode and the standing wave correspond to the anti-symmetric (A_0) Lamb wave, which has been considered as optimal for piezoelectric haptics in previous works¹².

We think that a study of dispersion curves of Lamb waves is beyond the scope of this article. However, Lamb waves in glass plates for haptics were extensively studied by Bernard¹³. Below we show a Figure from his work showing wavevectors of Lamb waves as functions of a frequency-thickness product. In our work the glass is 300 μm -thick and is operated at a frequency of 40.2 kHz, leading to the frequency-thickness product of 0.012 MHz mm. This is below the appearance of any other modes than A_0 . The fact that Bernard used EAGLE XG and we are using AF32 glass does not change the outcome of this analysis as both glasses have comparable mechanical properties and density.

[REDACTED]

Figure 5: **Wavevectors of different Lamb wave modes as function of frequency-thickness product.** Adapted from Bernard¹³.

We added the above Figure and corresponding text to the Supplementary section 3.2.3 “Device performance”.

In the main manuscript we modified part of the paragraph that describes Figure 4 (page 9) to: “At 40.2 kHz the device reveals a peak in deflection, which corresponds to its mechanical resonance at anti-symmetric (A_0) Lamb mode. For further info on Lamb mode analysis see supplementary Fig. S14 and Ref.⁴⁰”.

Comment 2.3

The title, “Low-temperature growth of piezoceramic films on glass” is too broad for this paper, which should be more specific. Relevant key words, such as flash lamp, and/or the material, PZT, should be added in the title.

Response 2.3

We thank the reviewer for this suggestion. The title has been changed to “Crystallization of piezoceramic films on glass via flash lamp annealing”.

Comment 2.4

Why the authors select Pt instead of a transparent electrode on the top, although interdigital electrode configuration is used?

Response 2.4

Our primary objective was to investigate the direct growth of PZT films onto glass substrates. We opted for Pt as it is the most studied and understood electrode material for piezoelectric thin films. Transparent electrodes, like indium tin oxide (ITO), do have transparency benefit, however, they can also bring extrinsic effects to the response, such as large resistance and non-inert interface between the electrode and the piezoelectric. As a side note, we do agree that a combination of transparent electrode with FLA-processed PZT is of high interest and are currently developing the process on that.

We added the following statement in Methods (Section 4.4) part of the manuscript: “Pt was chosen as it is the most mature electrode material for piezoelectric thin films.”

Comment 2.5

In Page 6, Lines 117-118, the “Further details on processing are given in Supplementary Section 2”, should be changed into “Section 4.1 Processing and deposition of PZT solution”, as the details on processing are not mentioned in the Supplementary Section 2.

Response 2.5

We apologize for this mistake. Reading the revised manuscript, we found this sentence redundant and have removed it.

Comment 2.6

In Pages 8-9, the content for comparing various annealing processes including the comparison table could be moved from the section titled, “Electromechanical Characterization” to the discussion section.

Response 2.6

We thank the reviewer for this suggestion. The text “We conducted a comparison of the electrical properties between 1 μm -thick flash lamp-processed films and RTA-processed films ...” was moved from the end of the “Electromechanical Characterization” section to the “Discussion” section.

Comment 2.7

In Page 10, Lines 208, 217 and 228 mentioned Supplementary Section 4. However, “Section 2.4 Device Application” is about the film on AF32 glass, which correspond to the Supplementary Section 3. Note Page 11, Line 237, where the Supplementary Section 5 should be Supplementary Section 4. The authors are reminded of checking and correcting citations for the corresponding supplementary sections.

Response 2.7

We apologize for these mistakes. We have re-checked and corrected the section numbers.

Comment 2.8

In Page 12, there mentioned two kinds of FLA process which include one-step (Line 272) and two-step (Line 257) process. The one-step process is introduced in details, but not for the two-step process, and it is not clear why the two-step process was used. Any advantages compared with one-step process?

Response 2.8

The one-step process used for fused silica and AF32 glass is not suitable for growing PZT film on soda lime glass due to appearance of cracks. This can be attributed to the low thermal conductivity ($1.0 \text{ W m}^{-1} \text{ K}^{-1}$) of the substrate, which leads to a slower rate of heat transfer and consequently a higher temperature at the interface between the film and glass.

To address this issue, we have developed a two-step process consisting of “nucleation” and “growth” stages. In the first step, 6 pulses with higher power density (2.5 J cm^{-2} and $170 \mu\text{s}$) are applied to induce the formation of nuclei within the film. This formation of nuclei reduces the activation energy required for the phase transition from an amorphous to a crystalline phase. In the second step, the “growth” step, 240 pulses with a lower power density (2.5 J cm^{-2} and $250 \mu\text{s}$) are applied to grow the film at a lower temperature, thereby preventing the occurrence of cracks. For both steps, the repetition rate is set to 0.5 Hz to increase the heat diffusion through the sample.

The text above has been added to the Supplementary Information as Section 4.1. In the main manuscript, we added the following sentence on page 10 (Section 2.5): “This process is described in Supplementary Information Section 4, where also more details on phase composition and electrical properties are described (Fig. S16 and Fig. S17).”

Comment 2.9

Figures 2 and 3 both show the PZT film on fused silica glass. However, Figure 2 shows the microstructure and optical characterization for the film of 170-nm in thickness, and Figure 3 provides electromechanical characterization results for 500-nm in thickness. Why do not provide the results for the same thicknesses?

Response 2.9

We thank the reviewer for this remark, and we draw their attention to the fact that similar comment was given by the reviewer 1 (Comment 1.9) and reviewer 3 (Comment 3.4). We made detailed response to these comments in Response 1.9.

Comment 2.10

In Supplementary Section 2, Page 8, Line 90, it mentioned “50 pulses exhibit the optimal electrical property”. In the main text, Page 6 Lines 125-126, “100 pulses indicating the enhanced crystallinity”, but not mentioned which pulses has optimal property. Should the point in the supplementary info be mentioned in the main text for clarity.

Response 2.10

We thank the reviewer for pointing out this inconsistency. We added the following sentence in the main manuscript, Section 2.3: “This section is focused on the films prepared with 50 pulses, as they show optimal ferroelectric response.”

Reviewer 3

The authors investigate the so-called low-temperature flash lamp process for the growth of piezoelectric lead zirconate titanate films (PZT), in which various types of glasses are used as substrates. The main limitation in processing PZT films on glass is applying temperatures higher than 700 °C for the effective annealing of the films while keeping the glass substrate at temperatures below 600 °C, which has been successfully overcome in this study. While the overall study is well designed, the data are meticulously analysed, and the interpretation and methodology are sound, I have reservations about the novelty of the findings and outlook. To my opinion they do not meet the standards of novelty and significance expected for publication in Nature Comm.

Response

We thank the reviewer for generally positive opinion about the work. Their reservation about the novelty and meeting the Nature Communication standards are addressed in the points below.

Comment 3.1

Significance: The central question addressed by the study is important, and the use of FLA is beneficial for high-throughput, and large-scale roll-to-roll production. However, the reported findings do not present novel insights or unexpected outcomes. The manuscript lacks the transformative impact that is expected from publications in high-impact journals.

Novelty and Specific Concerns: The research presented in this manuscript appears to be a continuation of previous studies related to the FLA of PZT films and does not introduce new concepts or approaches.

The unique aspects of the authors' work that differentiate it from the existing literature are poorly highlighted. As briefly mentioned in the manuscript, the presented concept is already available in the literature:

- DOI: 10.1143/JJAP.41.2630
- DOI: 10.1016/j.jeurceramsoc.2020.07.052
- DOI: 10.1002/adma.202303553
- DOI: 10.1111/jace.14272
- Thesis: Ouyang, Jing, "Enhanced Piezoelectric Performance of Printed PZT Films on Low Temperature Substrates" (2017)
- Thesis: Marotta, Amanda R., "Printable Thin-Film Sol-Gel Lead Zirconate Titanate (PZT) Deposition Using NanoJet and Inkjet Printing Methods" (2019).

Response 3.1

We acknowledge reviewer's criticism and the fact that the novelty has not been stressed clearly enough. The most significant result of this work is a PZT film with macroscopic piezoelectric and ferroelectric properties obtained via *in-situ* crystallization induced by FLA treatment. In previous works FLA treatment was performed on either already crystalline films (particles) or macroscopic properties could not be measured. To support our claim, on the top of our initial analysis performed in the first version of the paper, we reviewed the literature listed by the reviewer. Our analysis is displayed in the Table 3 below.

Table 3: Summary of relevant points of the references listed by reviewer 3 and major advancement shown in this work.

Reference	Relevant points in the reference	Major advancement in our work compared to reference
Yamakawa et al., Jpn. J. Appl. Phys., 41 2630 (2002) ¹⁴ .	 • FLA treatment of PZT thin films. • Ambient temperature between 300 and 500 °C. • Crystalline perovskite phase present prior FLA treatment. • No piezoelectric properties. 	 • Ambient environment at room temperature. • Crystallization of completely amorphous initial films. • Demonstration of piezoelectric properties and a device.
Yao et al., J. Eur. Ceram. Soc., 40, 5396 (2020) ¹⁵ .	 • FLA treatment of PZT thin films. • No macroscopic electromechanical characterization (films too leaky). 	 • Demonstration of macroscopic electromechanical properties and a device.
Palneedi et al., Adv. Mater., 2303553 (2023) ¹⁶ .	 • FLA sintering of crystalline PZT powders deposited on metglas (amorphous metal). 	 • In-situ FLA crystallization of amorphous PZT thin films.
Ouyang et al., J. Am. Ceram. Soc., 99, 2569 (2016) ¹⁷ .	 • FLA sintering of crystalline PZT powders on stainless steel. • Non-saturated P-E loops. 	 • In-situ FLA crystallization of amorphous PZT thin films. • Good ferroelectric properties.
Ouyang, PhD Thesis, Rochester Institute of Technology (2017) ¹⁸ .	 • FLA sintering crystalline PZT powders on stainless steel and PET substrates. • Non-saturated P-E loops. 	 • In-situ FLA crystallization of amorphous PZT thin films. • Good ferroelectric properties.
Marotta, MSc Thesis, Rochester Institute of Technology (2019) ¹⁹ .	 • FLA treatment of printed PZT thin films. • No crystallization (XRD) reported. • No macroscopic electromechanical characterization. 	 • In-situ FLA crystallization of amorphous PZT thin films. • Demonstration of macroscopic electromechanical properties.

Two major points are stemming from the Table:

- 1) In all the previous reports with demonstrated macroscopic functional properties, FLA sintering was performed on already crystalline PZT powders. We performed FLA **crystallization from amorphous films**, i.e., nucleation of perovskite grains and their growth (see Figure 6). Perovskite formation is nucleation-controlled, with activations energies for nucleation and grain growth of 441 kJ mol⁻¹ and 112 kJ mol⁻¹, respectively²⁰. In other words, the most energetically demanding process for perovskite formation is nucleation from the amorphous phase, which our study was the only one to demonstrate.

- 2) In the remaining reports, where they worked on FLA treatment of amorphous PZT films, macroscopic ferroelectric results could not be obtained. The only exception is the work of Yamakawa et al.¹⁴ on sputtered PZT films. But note that they did FLA treatment at elevated ambient temperatures, and XRD reveals the presence of the perovskite phase already before FLA treatment (Figure 8 in their article). Therefore, our work is the only one that shows **macroscopic ferroelectric results** starting from fully amorphous films.

As a conclusion, our analysis shows that our work is the only one that reports macroscopic ferroelectric results on films that were fully amorphous before Flash Lamp Annealing.

Figure 6: **XRD study**. GIXRD pattern of an as-deposited (before FLA treatment) PZT film on fused silica glass. Peaks of the perovskite phase are completely absent.

Comment 3.2

There have already been reports on PZT films spin coated and printed from the ink-based solution, annealed with FLA on glass and plastic substrates. There is no clear evidence if the ink formulation, FLA processing parameters, or deposition methods in the manuscript are conceptually different compared with the reported results.

Response 3.2

The reviewer states a valid concern, and we acknowledge that we should have stressed better the **uniqueness** of our approach. What differentiates our FLA process from the ones previously reported is the unique tool design, which enables creation of **high-power pulses** (above 20 kW cm⁻²) with **high pulse repetition rates** (3 Hz). This process delivers enough energy to

amorphous film to initiate the crystallization (nucleation) process resulting in final films with good functional response. This claim is supported by the literature analysis displayed in the Table below.

Table 4: FLA processing parameters taken from the references cited by reviewer 3 and compared to the parameters used in this work. Note that Ouyang’s PhD thesis¹⁸ is omitted as its results are summarized in the article in Journal of the American Ceramics Society¹⁷. Marotta’s MSc thesis¹⁹ is omitted from this analysis also, as little information on FLA process is given. *Energy per pulse in Yamakawa’s work¹⁴ is estimated from the current delivered to the Xe lamp when the voltage is applied. Real energy delivered to the sample is probably much lower. +Power per pulse (unless given) is estimated from the energy divide by pulse width. #In this work we measured the energy delivered to the sample using bolometer.

Reference	Energy per pulse J cm ⁻²	Pulse width μs	Power per pulse ⁺ kW cm ⁻²	Number of pulses	Repetition rate Hz
Yamakawa et al., Jpn. J. Appl. Phys., 41 2630 (2002) ¹⁴ .	27*	1000-1500	18-27	up to 5	not given
Yao et al., J. Eur. Ceram. Soc., 40, 5396 (2020) ¹⁵ .	not given	250-500	Up to 6.4	up to 100	not given
Palneedi et al., Adv. Mater., 2303553 (2023) ¹⁶ .	1.7 – 7.4	250-1000	7	up to 3	1
Ouyang et al., J. Am. Ceram. Soc., 99, 2569 (2016) ¹⁷ .	2.8	1300	2.2	Up to 15	2
This work	3 [#]	130	up to 23	Up to 100	3

Following comments 3.1 and 3.2 we did the following modifications in the manuscript:

Supplementary Information. We added Section 5 “Comparison with previous works” in which we placed the above tables and corresponding text.

Main paper. We added a paragraph in the discussion: “Macroscopic ferroelectric and piezoelectric properties of the films crystallized during flash lamp annealing are demonstrated in this work. The previously unreported process of nucleation and grain growth of the perovskite phase has been enabled by the unique tool design, which enables creation of high-power pulses (above 20 kW cm⁻²) with high pulse repetition rates (3 Hz)^{30,42–44}. Detailed comparison with the literature can be found in Supplementary Section 5.”

Note that we also added the references Yamakawa et al., Palneedi et al. and Ouyang et al.

Comment 3.3

The conceptualization of the study as a “low-temperature growth” process is considered wrong, since the film is deposited and annealed at high temperatures, according to the simulations, while only the substrate remains at low temperatures.

Response 3.3

We conceptualized the “low-temperature growth” because the ambient environment remains at room temperature. Considering the above criticism and to better reflect the outcome of literature analysis, we changed the title to: “Crystallization of piezoceramic films on glass via flash lamp annealing”. Note that we also removed the term “low-temperature” in the text where relevant.

Comment 3.4

The variation of the film thickness from 170 nm to 500 nm and 1 μm for different types of substrates is confusing.

Response 3.4

We thank the reviewer for this remark, and we draw their attention to the fact that similar comment was given by the reviewer 1 (Comment 1.9) and 2 (Comment 2.9). We made detailed response to these comments in Response 1.9.

Recommendation

Based on the concerns outlined above, I suggest that the authors submit their manuscript to a more specialized journal that focuses on ceramic film processing. This would provide a more appropriate platform for the dissemination of their work to a targeted audience who would better appreciate the incremental advancements made by this study.

Response 3.5

We hope that with the replies and analyses above we managed to clearly demonstrate the novelty (previously inaccessible processing parameters of FLA process) and the significance (in-situ FLA crystallization in ambient environment) of this study. As the results pave the way towards efficient integration of piezoelectric devices on glass, we strongly believe that it possesses transformative nature for publication in Nature Communications.

Additional corrections

We would like to point out that we corrected the following points in addition to the reviewers' remarks:

- When describing flash lamp annealing parameters, the terms “frequency” and “pulse length” were replaced by “repetition rate” and “pulse duration”, respectively. This correction was done to improve accuracy of the manuscript.
- It was clarified in the caption of Figure 2 that all films used in that Figure are 170 nm-thick.
- In the Supplementary, the electrical field in the Figure S9a and b was corrected.
- In Section 3.1 of the Supplementary, the repetition rate was given along with the other flash lamp annealing parameters.
- The reference for the standard value of Young's modulus of PZT in Section 3 was replaced by a reference focusing on sol-gel thin film PZT²¹.
- In Table 1, the numbers of significant digits for $\epsilon_{33,f}$ values were uniformised (one digit) and the dielectric losses were given as absolute values instead of percentage.
- The numbering of the Sections and Paragraphs in the Supplementary was simplified into a section numbering (e.g., 2.2.1.).
- Figures in the Supplementary Information that were not introduced in the main article or in the Supplementary have been mentioned accordingly.
- An expression “several seconds per crystallization” was added on page 4.

References:

1. Sadl, M. *et al.* Energy-storage-efficient $0.9\text{Pb}(\text{Mg}_{1/3}\text{Nb}_{2/3})\text{O}_3$ - 0.1PbTiO_3 thick films integrated directly onto stainless steel. *Acta Mater.* **221**, 117403 (2021).
2. Patil, D. R. *et al.* Piezoelectric thick film deposition via powder/granule spray in vacuum: A review. *Actuators* **9**, 59 (2020).
3. Hwang, G. T. *et al.* Self-powered wireless sensor node enabled by an aerosol-deposited PZT flexible energy harvester. *Adv. Energy Mater.* **6**, 1–9 (2016).
4. Song, L., Glinsek, S. & Defay, E. Toward low-temperature processing of lead zirconate titanate thin films: Advances, strategies, and applications. *Appl. Phys. Rev.* **8**, 041315 (2021).
5. Sadl, M. *et al.* Multifunctional energy storage and piezoelectric properties of $0.65\text{Pb}(\text{Mg}_{1/3}\text{Nb}_{2/3})\text{O}_3$ - 0.35PbTiO_3 thick films on stainless-steel substrates. *J. Phys. Energy* **4**, 0–9 (2022).
6. ICDD database PDF4+ v.19. (2019).
7. Hofmann, A. I., Cloutet, E. & Hadziioannou, G. Materials for transparent electrodes: from metal oxides to organic alternatives. *Adv. Electron. Mater.* **4**, 1700412 (2018).
8. Murali, P. Recent progress in materials issues for piezoelectric MEMS. *J. Am. Ceram. Soc.* **91**, 1385–1396 (2008).
9. Glinsek, S. *et al.* Fully transparent friction-modulation haptic device based on piezoelectric thin film. *Adv. Funct. Mater.* **30**, 2003539 (2020).
10. Glinsek, S. *et al.* Inkjet-printed piezoelectric thin films for transparent haptics. *Adv. Mater. Technol.* **7**, 2200147 (2022).
11. Hua, H., Chen, Y., Tao, Y., Qi, D. & Li, Y. A highly transparent haptic device with an extremely low driving voltage based on piezoelectric PZT films on glass. *Sensors Actuators A Phys.* **335**, 113396 (2022).
12. Bernard, F., Casset, F., Danel, J. S., Chappaz, C. & Basrour, S. Characterization of a smartphone size haptic rendering system based on thin-film AlN actuators on glass substrates. *J. Micromechanics Microengineering* **26**, 84007 (2016).
13. Bernard, F. Conception, fabrication et caractérisation d'une dalle haptique à base de

- microactionneurs piézoélectriques. (Université Grenoble Alpes, 2016).
14. Yamakawa, K. *et al.* Novel Pb(Ti,Zr)O₃ (PZT) crystallization technique using flash lamp for ferroelectric RAM (FeRAM) embedded LSIs and one transistor type FeRAM devices. *Jpn. J. Appl. Phys.* **41**, 2630–2634 (2002).
 15. Yao, Y. *et al.* Direct processing of PbZr_{0.53}Ti_{0.47}O₃ films on glass and polymeric substrates. *J. Eur. Ceram. Soc.* **40**, 5369–5375 (2020).
 16. Palneedi, H. *et al.* Intense pulsed light thermal treatment of Pb(Zr,Ti)O₃/metglas heterostructured films resulting in extreme magnetoelectric coupling of over 20 V cm⁻¹ O⁻¹. *Adv. Mater.* 2303553 (2023).
 17. Ouyang, J., Cormier, D., Williams, S. A. & Borkholder, D. A. Photonic sintering of aerosol jet printed lead zirconate titanate (PZT) thick films. *J. Am. Ceram. Soc.* **99**, 2569–2577 (2016).
 18. Ouyang, J. Enhanced piezoelectric performance of printed PZT films on low temperature substrates. (Rochester Institute of Technology, 2017).
 19. Marotta, A. R. Printable thin-film sol-gel lead zirconate titanate (PZT) deposition using nanojet and inkjet printing methods. (Rochester Institute of Technology, 2019).
 20. Chen, K. C. & Mackenzie, J. D. Crystallization kinetics of metallo-organics derived PZT thin film. *MRS Online Proc. Libr.* **180**, 663–668 (1990).
 21. Casset, F. *et al.* Young modulus and Poisson ratio of PZT thin film by picosecond ultrasonics. in *IEEE International Ultrasonics Symposium, IUS* 2180–2183 (2012).

Reviewers' comments:

Reviewer #2 (Remarks to the Author):

The authors have well responded to most of my comments with corrections and clarifications. The quality of the manuscript has been improved substantially after the revisions.

Since the haptic device is the most relevant and important application for reflecting the value of the low processing piezoelectric film on transparent glass substrate, an improved connection between the actuator specifications and requirements of haptic sensing is necessary, but not clear in this work with most literature citation on materials. A straightforward way to enhance the manuscript for broad readership could be to cite state-of-the-art review paper or latest book clearly stating or summarizing the required performance of electromechanical actuators for haptic sensing feedback applications.

Reviewer #3 (Remarks to the Author):

The author have prepared an extensive reply and also modified the title, which is highly appreciated. Nevertheless, my main concern about the missing novelty and originality remains.

Tables 3 and 4 provided in the Response letter justify the numerous literature available on FLA of PZT, including two studies on FLA of amorphous PZT precursors. Reference [30] Yao et al. reports identical experiments on FLA of amorphous PZT precursors. From a broader perspective, there are dozens of studies on the low-temperature sintering of amorphous PZT precursors films using thermally- and photon-assisted methods, which is illustrated in Fig. 6 of this review [Applied Physics Reviews 8, 041315 (2021) <https://doi.org/10.1063/5.0054004>].

To my opinion, provided Tables 3 and 4 just emphasize that the present study presents only incremental improvements as compared to the numerous state-of-the-art. The manuscript does not report any new concept, approach or important experimental finding. It does not represent any major advances of significance to specialists working in FLA or PZT.

Recommendation:

Based on the remaining concern outlined above, I suggest that the authors submit their manuscript to a more specialized journal that focuses on ceramic film processing. This would provide a more appropriate platform for the dissemination of their work to a targeted audience who would better appreciate the incremental technological advancements made by this study.

Crystallization of piezoceramic films on glass via flash lamp annealing

Submission ID: NCOMMS-23-20861

We thank both reviewers again for their valuable comments. We considered them and implemented changes in the manuscript accordingly (highlighted in yellow). The comments are addressed in details hereafter, point by point.

Reviewer 2

The authors have well responded to most of my comments with corrections and clarifications. The quality of the manuscript has been improved substantially after the revisions.

Response:

We would like to thank the reviewer for their positive feedback.

Comment 2.1

Since the haptic device is the most relevant and important application for reflecting the value of the low processing piezoelectric film on transparent glass substrate, an improved connection between the actuator specifications and requirements of haptic sensing is necessary, but not clear in this work with most literature citation on materials. A straightforward way to enhance the manuscript for broad readership could be to cite state-of-the-art review paper or latest book clearly stating or summarizing the required performance of electromechanical actuators for haptic sensing feedback applications.

Response 2.1

We thank the reviewer for this comment, which indeed is very relevant for general audience's understanding of the technology. The haptic device demonstrated in this work is based on ultrasonically vibrating surface, which can modulate forces at the interface between a finger

and a vibrating plate. The so-called friction-modulation effect is applicable in touchscreen with foreseen applications such as displays for automotive industry, displays for visually disabled people, or sliders for control of heating/cooling¹. The effect depends strongly on the out-of-plane displacement, in-plane wavelength, and frequency of the standing acoustic wave, which is created in the screen. For technology commercialization, the following criteria have to be fulfilled: amplitude must be larger than 1 μm (enabling detection with a human finger and significant decrease of a friction coefficient)^{2,3}, wavelength must be below ~ 15 mm (enabling detection with a human finger)⁴ and its frequency should be beyond 25 kHz (enabling silent operation)⁵. Note that all the criteria have been fulfilled by the demonstrated device.

Beyond these properties, it is important to mention that in hand-held electronics it is easier to handle large voltage-values compared to large current-values, if the voltage remains below 100 V⁶. 60 V needed to drive demonstrated device can be achieved by using ASICs or by making an LC resonator. Note that this is already discussed in the paper.

To make this point clearer in the main manuscript and to follow reviewer's advice, we added the following paragraph in its introduction:

“Piezoelectrics have been recently demonstrated as efficient actuators for haptics. The technology is based on ultrasonically vibrating surface, which can modulate forces at the interface between a finger and a vibrating plate. The so-called friction-modulation effect is applicable in touchscreens with foreseen applications such as displays for automotive industry, displays for visually disabled people, or sliders for control of heating/cooling³¹. The effect depends strongly on the out-of-plane displacement, in-plane wavelength, and frequency of the standing acoustic wave, which is created in the screen. For technology commercialization, the following criteria have to be fulfilled: amplitude must be larger than 1 μm (enabling detection with a human finger and significant decrease of a friction coefficient)^{32,33}, wavelength must be below ~ 15 mm (enabling detection with a human finger)³⁴ and its frequency should be beyond 25 kHz (enabling silent operation)³⁵.”

In addition to the above text, we added the following references: a review paper by Basdogan et al. (Ref. ³¹) and 4 additional papers that are important in the field (Ref. ³²⁻³⁵).

Reviewer 3

The author have prepared an extensive reply and also modified the title, which is highly appreciated. Nevertheless, my main concern about the missing novelty and originality remains.

Response

We thank the reviewer for their appreciation of the effort. We address further the novelty and the originality below.

Comment 3.1

Tables 3 and 4 provided in the Response letter justify the numerous literature available on FLA of PZT, including two studies on FLA of amorphous PZT precursors. Reference [30] Yao et al. reports identical experiments on FLA of amoprphous PZT precursors. From a broader perspective, there are dozens of studies on the low-temperature sintering of amorphous PZT precursors films using thermally- and photon-assisted methods, which is illustrated in Fig. 6 of this review [Applied Physics Reviews 8, 041315 (2021) <https://doi.org/10.1063/5.0054004>].

Response 3.1

We address this comment point by point below.

Studies on FLA of amorphous PZT precursors. The two studies focusing on FLA of amorphous PZT precursors are the Marotta's MSc Thesis⁷ and Yao et al.'s paper⁸. **Marotta reports neither crystallisation (absence of XRD analysis) nor macroscopic electromechanical characterization⁷.** Regarding Yao et al.'s work⁸, the statement that the experiments are identical to our own is inaccurate. Two key differences exist compared to our work:

- Electromechanical properties. We demonstrate macroscopic ferroelectric and piezoelectric properties and we realize a piezoelectric device. **In Yao's work, macroscopic properties could not be measured (claim in their paper), therefore their films cannot be used in applications.**
- FLA process. The advancement of our work was enabled by a significantly different process. Ours is based on high-energy and short-width pulses, which lead to the power of 15 kW cm⁻² per pulse. Application of such high powers was enabled by a unique tool that we designed and led to creation of grains with the size of ~100 nm (see Figure S6 in our paper). Yao et al.'s work, on the other hand, is based on three times lower-power (~6.4 kW cm⁻² per pulse), which is

not enough to perform full crystallization and grain growth resulting in the nano-sized grains (see Figure 3 in their paper).

Studies on low temperature sintering of amorphous PZT. The review by Song, Glinsek and Defay (note that we are also co-authors of this paper) covers low-temperature processing of lead zirconate titanate thin films⁹. In Figure 6 of that paper, we compared values of remanent polarization P_r achieved by various low temperature techniques. We therefore agree with the reviewer that there are many reports on low-T processes. But note that we used P_r as the comparison property because piezoelectric properties of low-T films are only seldomly reported^{10,11}. High P_r values are not synonymous to high piezoelectric coefficient as they can be poised by leakage current contribution. In our current work we advance further as we, in addition to evaluation of piezoelectric effect, **present a first-ever piezoelectric device based on piezoelectric films grown at low-temperature (being thermal or photonic-assisted process).**

Comment 3.2

To my opinion, provided Tables 3 and 4 just emphasize that the present study presents only incremental improvements as compared to the numerous state-of-the-art. The manuscript does not report any new concept, approach or important experimental finding. It does not represent any major advances of significance to specialists working in FLA or PZT.

Response 3.2

We respectfully disagree with this statement of the reviewer.

The major advancement of this work is a first-ever piezoelectric device based on piezoelectric films grown at low-temperature (being thermal or photonic-assisted process). This advancement was achieved by a unique tool design, which enables creation of high-power pulses (above 15 kW cm⁻²) with high pulse repetition rates (3 Hz). Our work presents a new concept with the use of a unique, customised FLA tool (high power at high frequency) and annealing strategies developed for PZT on various glass substrates. Furthermore, this work demonstrates macroscopic piezoelectric properties and a first-ever piezoelectric device based on films grown at low temperature.

Recommendation

Based on the remaining concern outlined above, I suggest that the authors submit their manuscript to a more specialized journal that focuses on ceramic film processing. This would provide a more appropriate platform for the dissemination of their work to a targeted audience who would better appreciate the incremental technological advancements made by this study.

Response 3.3

As this paper demonstrates functional piezoelectric device obtained via innovative processing, we think its content is too broad for a specialized journal on ceramic processing. As the process offer a potential for large-scale processing ¹², we also strongly believe that it possesses transformative nature for publication in Nature Communications.

Additional corrections

We would like to point out that in addition to the reviewers' remarks we corrected the scale of the x-axis in Figure 3b. Originally ranging between -1 and 1 MV cm⁻¹, it now ranges between -250 and 250 kV cm⁻¹. Note that this does not change the results and their analysis.

We also removed Reference 31 (Kim2015) from the paper as it became redundant with those added in the corrections.

References:

1. Basdogan, C., Giraud, F., Levesque, V. & Choi, S. A Review of Surface Haptics: Enabling Tactile Effects on Touch Surfaces. *IEEE Trans. Haptics* **13**, 450–470 (2020).
2. Bernard, F., Gorisse, M., Casset, F., Chappaz, C. & Basrour, S. Design, fabrication and characterization of a tactile display based on aln transducers. *Procedia Eng.* **87**, 1310–1313 (2014).
3. Wiertelwski, M., Friesen, R. F. & Colgate, J. E. Partial squeeze film levitation modulates fingertip friction. *Proc. Natl. Acad. Sci. U. S. A.* **113**, 9210–9215 (2016).
4. Bernard, F. Conception, fabrication et caractérisation d'une dalle haptique à base de microactionneurs piézoélectriques. (Université Grenoble Alpes, 2016).

5. Sednaoui, T. *et al.* Experimental Evaluation of Friction Reduction in Ultrasonic Devices. *IEEE World Haptics Conf. WHC 2015* 37–42 (2015) doi:10.1109/WHC.2015.7177688.
6. Glinsek, S. *et al.* Fully Transparent Friction-Modulation Haptic Device Based on Piezoelectric Thin Film. *Adv. Funct. Mater.* **30**, 2003539 (2020).
7. Marotta, A. R. Printable thin-film sol-gel lead zirconate titanate (PZT) deposition using nanojet and inkjet printing methods. (Rochester Institute of Technology, 2019).
8. Yao, Y. *et al.* Direct processing of $\text{PbZr}_{0.53}\text{Ti}_{0.47}\text{O}_3$ films on glass and polymeric substrates. *J. Eur. Ceram. Soc.* **40**, 5369–5375 (2020).
9. Song, L., Glinsek, S. & Defay, E. Toward low-temperature processing of lead zirconate titanate thin films: Advances, strategies, and applications. *Appl. Phys. Rev.* **8**, 041315 (2021).
10. Fink, S., Lübben, J., Schneller, T., Vedder, C. & Böttger, U. Impact of the processing temperature on the laser-based crystallization of chemical solution deposited lead zirconate titanate thin films on short timescales. *J. Appl. Phys.* **131**, 125302 (2022).
11. Kang, M. G. *et al.* Direct growth of ferroelectric oxide thin films on polymers through laser-induced low-temperature liquid-phase crystallization. *Chem. Mater.* **32**, 6483–6493 (2020).
12. Loganathan, K. *et al.* Rapid and up-scalable manufacturing of gigahertz nanogap diodes. *Nat. Commun.* 2022 **13**, 3260 (2022).

REVIEWER COMMENTS

Reviewer #3 (Remarks to the Author):

The authors have prepared another extensive reply. Nevertheless, my main concern about missing novelty and originality remains.

It is stated in the reply "The major advancement of this work is a first-ever piezoelectric device based on piezoelectric films grown at low-temperature (being thermal or photonic-assisted process)". I disagree with this "advancement" because of the following arguments:

1) It is certainly NOT the "first-ever piezoelectric device based on piezoelectric films grown at low-temperature" because there are dozens other approaches not using FLA. The search in WebOfScience for keywords "PZT" + "piezoelectric device" + "low-temperature" + "thin-film" returns 40 (!) results.

2) There are numerous alternative fabrication approaches like PLD <https://publikationen.bibliothek.kit.edu/1000072020> - low temperature (445 °C) or - thin-film PZT on polymer foil by lift-off <https://onlinelibrary.wiley.com/doi/full/10.1002/adma.201305659>

3) Apart from PZT there are many other piezoelectric material materials with low-temperature processing, e.g. perovskites <https://link.springer.com/article/10.1007/s12274-019-2486-5> AlN <https://pubs.aip.org/aip/apl/article/36/8/643/527037/Low-temperature-growth-of-piezoelectric-AlN-film>

4) Lastly, FLA is NOT a low-temperature process. FLA implies film heating to temperatures 700, 800, 1000 °C and above. The substrate bulk remains much colder because of the limited pulse duration and the rapid cooling via emission and heat dissipation into substrate. FLA fundamentals can be found in numerous textbooks, e.g. <https://link.springer.com/book/10.1007/978-3-030-23299-3>. Claim "low-temperature" is not justified unless one provides temperature values, either measured or simulated.

The other claims are highly arguable (or false), too:

"This advancement was achieved by a unique tool design, which enables the creation of high-power pulses (above 15 kW cm⁻²) with high pulse repetition rates (3 Hz)". It is not a unique tool, numerous groups worldwide use PulseForge tools offering > 40 kW/cm² and rep rate in kHz regime.

"Our work presents a new concept with the use of a unique, customised FLA tool (high power at high frequency) and annealing strategies developed for PZT on various glass substrates" The concept of using FLA for annealing of PZT on glass is not new (see numerous refs in both re-views).

"Furthermore, this work demonstrates macroscopic piezoelectric properties and a first-ever piezoelectric device based on films grown at low temperature". ALL the ferroelectric materials exhibit piezoelectric effect due to lack of symmetry. Therefore, PZT films with well-characterized ferroelectric

properties by Yao et al. do CONFIRM the films are piezoelectric. Other references <https://onlinelibrary.wiley.com/doi/full/10.1002/adma.202303553>, <https://ceramics.onlinelibrary.wiley.com/doi/full/10.1111/jace.14272> also describe piezoelectric properties of FLA-processed PZT, also the two mentioned PhD theses. FLA companies even re-port these in their product advertisements!
https://www.ulvac.co.jp/en/news/ulvac_launches_revolutionary_pzt_piezoelectric_thin-film_process_technology_and_hvm_solution_for_mem/

Reviewer #4 (Remarks to the Author):

In general, the paper deals with an interesting topic worth of publishing. However, there are few weaknesses in presenting the data and a minor methodical mistake, which is why I recommend revision. In detail

(1) The splitting between the article and the supplemental information is disadvantageous. There is a lot of information in the supplemental information which are closely intertwined with that of the article, and the reader is frequently forced to switch between both. I suggest a better content-related separation of both documents on the premise that the supplemental information is only a support, not an essential part of the paper. Sometimes there is a redundancy, e.g. between Fig. 2b (transmittance for 170 nm) and Fig. S5 (transmittance for 170 nm, 500 nm, 1 μ m).

(2) The authors used two methods to determine the absorptance of an amorphous PZT layer, but both are somewhat insufficient as they neglect significant contribution. The bolometer correctly measures transmittance, but neglect the reflected part of the light. Thus, this method significantly overestimates the absorptance. To measure transmittance and reflectance is the right way, but the final step of this method is missing: the mathematical convolution (or weighted average) of curve A in fig. S1b with the flash lamp spectrum on the energy scale. This would consider the “non-uniform absorbance of the sample through the relevant spectrum range”, and the authors could use a more correct absorbance value. If not measured, the flash lamp spectrum can be obtained by Novacentrix (see 4.2.) or alternatively under https://www.heraeus.com/media/media/hng/doc_hng/products_and_solutions_1/arc_and_flash_lamps_1/6kW_flaslamp_module_EN.pdf

(3) Please check the numbers in chapter 4.1. of the supplemental; for both high and low power densities an energy density of 2.5 Jcm⁻² is given. Furthermore, I doubt that you can separate nucleation and crystal growth so sharply, after exactly 6 pulses. It would be better to speak about 2 phases where nucleation or crystal growth is dominating.

Reviewer #5 (Remarks to the Author):

This manuscript entitled 'Crystallization of piezoceramic films on glass via flash lamp annealing' reports a flash lamp crystallization of PZT film on glass substrate. The annealed PZT film has a reasonable ferroelectric, piezoelectric, and transparent properties for various applications. Especially, the flash lamp annealing of solution-based PZT film on glass seems to have a novelty because the initial nature of solution coated PZT film is totally different with the physically coated films by sputtering and aerosol deposition. Additionally, the authors have effectively responded to the reviewers' comments through the revision process. The revised manuscript could be accepted in Nature Communication with current form.

Crystallization of piezoceramic films on glass via flash lamp annealing

Submission ID: NCOMMS-23-20861B-Z

We thank all the reviewers for additional valuable comments. We considered them and implemented changes in the manuscript accordingly (highlighted in yellow). The comments are addressed in details hereafter, point by point.

Reviewer 3

The authors have prepared another extensive reply. Nevertheless, my main concern about missing novelty and originality remains.

It is stated in the reply "The major advancement of this work is a first-ever piezoelectric device based on piezoelectric films grown at low-temperature (being thermal or photonic-assisted process)". I disagree with this "advancement" because of the following arguments:

Response:

We thank the reviewer for their time evaluating the manuscript. Our responses to their criticism are listed point by point below.

Comment 3.1

It is certainly NOT the "first-ever piezoelectric device based on piezoelectric films grown at low-temperature" because there are dozens other approaches not using FLA. The search in WebOfScience for keywords "PZT" + "piezoelectric device" + "low-temperature" + "thin-film" returns 40 (!) results.

Response 3.1

We would like to start the reply with a clear statement that we have already removed the term "low-temperature process" from the manuscript during previous round of reviews. Following

reviewer's criticism, we did the analysis on Web of Science, using the same terms as the reviewer. As a guidance in our analysis of the results, we used the temperature of 400 °C as a threshold value for a "low-temperature process", discussed by Bretos et al. ¹.

- 9 out of 40 papers are reviews and were not considered further.
- 29 out of remaining 31 papers are either works on bulk materials, at temperatures above 600 °C or they do not demonstrate piezoelectric device.
- The two remaining relevant papers are discussed here. Tue et al. ² successfully integrated PZT thin-film actuators on oxide TFTs at 450 °C. The work significantly deviates from ours as PZT is grown on Si substrates. Ueda et al. ³, on the other hand, reported transparent PZT films on FTO/glass substrates, which were processed at 550 °C. While excellent results were achieved, the processing temperature significantly exceeds 400 °C.

This analysis gives us further confirmation about the novelty of our work.

Comment 3.2

There are numerous alternative fabrication approaches like PLD <https://publikationen.bibliothek.kit.edu/1000072020> - low temperature (445 °C) or - thin-film PZT on polymer foil by lift-off <https://onlinelibrary.wiley.com/doi/full/10.1002/adma.201305659>

Response 3.2

Indeed, PZT films can be grown at rather low temperatures using pulsed laser deposition (PLD). However, PLD processing is performed in vacuum and it is difficult to scale-up ⁴. As discussed in the paper cited by the reviewer, low pressures (0.05 mbar) are needed to process the films at low temperatures ⁵. This deviates further from our ambient-pressure process.

Different lift-off processes have recently received a lot of attention. In these cases, the films are typically grown at high temperatures on appropriate substrates and are then transferred to temperature-sensitive substrates using sacrificial layers, laser lift-off, etc ^{6,7}. Indeed, this is an exciting approach, however, compared to direct growth it adds significant complexity to the fabrication process. We already mention this in the main paper, page 4: "The indirect

integration on glass via transfer process has been successful ²⁴, however, it significantly complicates the process and adds processing steps.”

Low-temperature PLD process is now mentioned in the manuscript on page 4, paragraph 1: “Pulsed laser deposition (PLD) method can lead to good films at 445 °C, however, low pressures (0.05 mbar) are required, and the method is difficult for scale-up ^{22,23}.”

Comment 3.3

Apart from PZT there are many other piezoelectric material materials with low-temperature processing, e.g. perovskites <https://link.springer.com/article/10.1007/s12274-019-2486-5> AlN <https://pubs.aip.org/aip/apl/article/36/8/643/527037/Low-temperature-growth-of-piezoelectric-AlN-film>

Response 3.3

The first reference that the reviewer cites ⁸ is an example of sputtered (K,Na)NbO₃ (KNN) thin films. The growth temperature of 350 °C reported in this paper is impressive. However, their P-E loops resemble leaky dielectric rather than ferroelectric character of these films. For comparison, see Figure 1a in a viewpoint by Prof. Scott ⁹, where the issue of lossy dielectrics is discussed.

We do agree with the reviewer that other technologically important piezoelectric materials, whose processing temperatures are lower compared to PZT (and oxide piezoelectrics in general), exist. AlN is indeed one of them, piezoelectric polymers (polyvinylidene difluoride (PVDF), etc.) are another example. However, piezoelectric coefficients of these materials are much lower ($e_{31,f}$ of ~ -3 C m⁻² and ~ -0.04 C m⁻² for AlN- and PVDF-based materials ^{10,11}). While their properties are appropriate for certain applications (resonators and energy harvesters for AlN and PVDF, respectively), they cannot completely replace PZT or other perovskites.

To reflect this point in the main manuscript, we added the following text in the first paragraph of the Introduction: “The two last ones are excellent materials for resonators and energy harvesters, respectively, and can be processed at low temperatures (e.g. <350 °C for AlN - and ~ 150 °C for polyvinyl difluoride (PVDF)-based materials). But much lower piezoelectric coefficients prevent them to replace perovskites, especially for actuator applications ⁶⁻⁸.”

Comment 3.4

Lastly, FLA is NOT a low-temperature process. FLA implies film heating to temperatures 700, 800, 1000 °C and above. The substrate bulk remains much colder because of the limited pulse duration and the rapid cooling via emission and heat dissipation into substrate. FLA fundamentals can be found in numerous textbooks, e.g. <https://link.springer.com/book/10.1007/978-3-030-23299-3>. Claim "low-temperature" Is not justified unless one provides temperature values, either measured or simulated.

Response 3.4

The reviewer pointed out this fact already in the previous round of reviews and we acknowledged it. Previous and current version of our manuscript does not use a term “low-temperature process” in relation with FLA processing.

Comment 3.5

The other claims are highly arguable (or false), too:

"This advancement was achieved by a unique tool design, which enables the creation of high-power pulses (above 15 kW cm⁻²) with high pulse repetition rates (3 Hz)". It is not a unique tool, numerous groups worldwide use PulseForge tools offering > 40 kW/cm² and rep rate in kHz regime.

Response 3.5

Our statement is based on a literature review we performed in the previous round of reviews (submitted on July 31st 2023, Response 3.2) and based on the feedback we got from the supplier. Our tool consists of 5 lamp drivers, which are connected to the flash lamp (75 x 150 mm²), which is a non-standard configuration for this tool. This enables high power with high repetition rate. We are not aware of a tool that can deliver > 40 kW/cm² with the repetition rate in kHz regime. To avoid ambiguity, we removed the term “unique” from the main paper.

Moreover, there is no report mentioning such power density combined with high pulse repetition rate after our thorough review of the state of the art.

Comment 3.6

"Our work presents a new concept with the use of a unique, customised FLA tool (high power at high frequency) and annealing strategies developed for PZT on various glass substrates" The concept of using FLA for annealing of PZT on glass is not new (see numerous refs in both reviews).

Response 3.6

We do agree with the statement that the concept of FLA annealing of PZT on glass is not new. But we do argue that we found a set of parameters for FLA annealing which leads to films with measurable macroscopic ferroelectric and piezoelectric properties and that we are able to demonstrate a device. We could not find any work in the literature, after a comprehensive review, approaching the performances we reached. And the parameters we used in FLA have not been disclosed in any known report. After three rounds of review, our state-of-the-art review has been very thorough.

Comment 3.7

"Furthermore, this work demonstrates macroscopic piezoelectric properties and a first-ever piezoelectric device based on films grown at low temperature". ALL the ferroelectric materials exhibit piezoelectric effect due to lack of symmetry. Therefore, PZT films with well-characterized ferroelectric properties by Yao et al. do CONFIRM the films are piezoelectric. Other references <https://onlinelibrary.wiley.com/doi/full/10.1002/adma.202303553>, <https://ceramics.onlinelibrary.wiley.com/doi/full/10.1111/jace.14272> also describe piezoelectric properties of FLA-processed PZT, also the two mentioned PhD theses. FLA companies even re-port these in their product advertisements! https://www.ulvac.co.jp/en/news/ulvac_launches_revolutionary_pzt_piezoelectric_thin-film_process_technology_and_hvm_solution_for_mem/

Response 3.7

Indeed, all ferroelectrics are piezoelectrics and we never claimed the opposite. What we do claim is that ferroelectricity observed with microscopic methods only is not a proof that

macroscopic piezoelectric devices can be realized. Below is a point-by-point comment to the references cited by the reviewer:

- Yao et al.¹². Indeed, these authors were successful in crystallizing amorphous PZT and prove the ferroelectricity via piezoresponse force microscopy (PFM). However, they pointed out in their paper they were not able to measure macroscopic properties.
- Palneedi et al., *Adv. Mater.*, 2023¹³. The authors report FLA treatment of already crystallized powders, which is inherently different process from ours. In addition, the paper appeared online after our initial submission to Nature Communications. Nonetheless, we have previously cited this article as Ref. 51 in the main manuscript and as Ref. 15 in the Supplementary Information.
- Ouyang et al.¹⁴. The authors report FLA treatment on already crystallized powders, which is inherently different process from ours.
- Reference to Ulvac's web-page. This is an advertisement for sputtered PZT films, and FLA process is not mentioned.

Hence, none of the cited papers reports on what we showed in this paper. We do not use already crystallized powder to prepare our films, which makes our process much more versatile.

As a conclusion to this new extensive answer to reviewer 3, we still believe that our work is unprecedented. To clarify this point in the main manuscript, we have added the following statement to the Introduction: "Hence, in this paper, we have shown that it is possible to manufacture a functional device (a haptic transducer) based on piezoelectric perovskite thin films deposited on a glass substrate and sintered with a specific Flash Lamp Annealing process. Moreover, we showed that a distinctive feature of the latter is that this crystallization can be performed on glass substrates that cannot withstand temperature larger than 400°C." and the statement "As a conclusion, we proved in this paper that a well-defined Flash Lamp Annealing process enables the crystallization of piezoelectric perovskite thin films deposited on glass substrates that cannot withstand temperatures larger than 400°C." to the Discussion section.

Reviewer 4

In general, the paper deals with an interesting topic worth of publishing. However, there are few weaknesses in presenting the data and a minor methodical mistake, which is why I recommend revision. In detail

Response

We thank the reviewer for their positive feedback.

Comment 4.1

The splitting between the article and the supplemental information is disadvantageous. There is a lot of information in the supplemental information which are closely intertwined with that of the article, and the reader is frequently forced to switch between both. I suggest a better content-related separation of both documents on the premise that the supplemental information is only a support, not an essential part of the paper. Sometimes there is a redundancy, e.g. between Fig. 2b (transmittance for 170 nm) and Fig. S5 (transmittance for 170 nm, 500 nm, 1 μm).

Response 4.1

We thank the reviewer for their advice, and we changed the splitting of the content between the article and the Supplementary Information. To facilitate the reading and remove redundancies, we made the following modifications:

- The redundancy of the transmittance curve of the 170 nm-thick PZT thin film on fused silica glass has been solved by removing this curve from the Supplementary Figure S4, as shown in Figure 1. This curve is now shown only in Figure 2 b) of the main article (see Figure 2 below).

Figure 1: **Transmittance of 500 nm and 1 μm -thick PZT thin films on fused silica glass.** Inset shows the optical appearance of the 1 μm -thick film. The films were processed with 50 pulses per layer (3 and 6 crystallizations in total, respectively). Parameters of each pulse are reported in caption of Fig. 2 in the main manuscript.

- The Supplementary Figure S3 shows a 170 nm-thick PZT thin film on fused silica substrate before annealing (0 pulses). It has been integrated to Figure 2a of the main article which shows the effect of various number of pulses used to anneal the same sample. The updated figure is shown in Figure 2.

Figure 2: **Microstructural and optical characterization of 170 nm-thick PZT films on fused silica glass.** a) GIXRD patterns of films annealed with different number of light pulses. In a) perovskite reflections are denoted according to a powder diffraction file (PDF) No 01-070-4264⁶⁴. Py indicates the pyrochlore reflection according to PDF No 04-014-5162⁶⁴. b) Transmittance of the 50-pulse film. Inset shows its optical appearance. STEM results of the 100-pulse film: c) cross sectional dark-field STEM image and EDS analysis across substrate/film interface (marked by dashed line) using Pb M and Si K line, d) EDS analysis across substrate-film interface using the Pb M and Si K lines showing partial diffusion of Si into the PZT film. The dashed area in c) marks the region where the d) EDS line analysis was performed, e) high-resolution bright-field STEM image of two perovskite grains with corresponding FFT image showing (110) planes. Flash lamp annealing was performed with energy density, pulse duration and repetition rate of 3 J cm⁻², 130 μs, and 3.5 Hz, respectively.

- The EDS analysis of a 170 nm-thick PZT thin film on fused silica substrate, originally introduced in the Supplementary Figure S8, has been integrated to Figure 2 of the main article. Indeed, the area of the EDS analysis is also represented in Figure 2 c) as part of a STEM analysis. The updated figure is shown in Figure 2.
- The parameters of the 2-step process for crystallization of PZT on soda-lime glass were originally described in the Supplementary Section 4.1. The following text has been added to Section 2.5 of the main article: “In the first step, 6 pulses with higher power density of 14.7 kW cm^{-2} (2.5 J cm^{-2} and $170 \mu\text{s}$) are applied. In the second step, 240 pulses with a lower power density of 10.0 kW cm^{-2} (2.5 J cm^{-2} and $250 \mu\text{s}$) are applied to grow the film at a lower temperature, thereby preventing the occurrence of cracks. For both steps, the repetition rate is set to 0.5 Hz to increase the heat diffusion through the sample.” The text in the Supplementary Section 4.1 has been modified to “In the first step, pulses with higher power density are applied to induce the formation of nuclei within the film. This formation of nuclei reduces the activation energy required for the phase transition from an amorphous to a crystalline phase. In the second step, the phase where crystal growth is dominating, pulses with a lower power density are applied to grow the film at a lower temperature, thereby preventing the occurrence of cracks.”

We have kept the studies on the PZT thin films thickness dependance and on the PZT thin films on soda lime glass in the Supplementary Information to keep the reading flow of the main article.

Comment 4.2

The authors used two methods to determine the absorptance of an amorphous PZT layer, but both are somewhat insufficient as they neglect significant contribution. The bolometer correctly measures transmittance, but neglect the reflected part of the light. Thus, this method significantly overestimates the absorptance.

To measure transmittance and reflectance is the right way, but the final step of this method is missing: the mathematical convolution (or weighted average) of curve A in fig. S1b with the flash lamp spectrum on the energy scale. This would consider the “non-uniform absorptance of

the sample through the relevant spectrum range”, and the authors could use a more correct absorbance value. If not measured, the flash lamp spectrum can be obtained by Novacentrix (see 4.2.) or alternatively under https://www.heraeus.com/media/media/hng/doc_hng/products_and_solutions_1/arc_and_flash_lamps_1/6kW_flaslamp_module_EN.pdf

Response 4.2

We thank the reviewer for their suggestion. While we did use both methods to estimate absorption in our samples, only the bolometer result was used for the modelling. Note that the same approach was recently used by the University of Texas / Novacentrix team in Ref. ¹⁵. We placed the bolometer in air (direct exposure to light, E_{tot}), below a bare glass substrate (estimation of light reflected from the glass substrate assuming negligible absorption, $E_{glass} = E_{tot} - E_{ref}$) and below a substrate with a film on top (estimation of light that passes through the stack, $E_{sample} = E_{trans}$). To estimate the quantity of absorbed light, we used the equation:

$$E_{abs} = E_{tot} - E_{sample} - E_{glass} = E_{tot} - E_{trans} - (E_{tot} - E_{ref}). \quad (1)$$

In the equation E denotes energy density in $J\ cm^{-2}$. The resulting E_{abs} is $0.8\ J\ cm^{-2}$. The absorption value A is estimated as:

$$A = \frac{E_{abs}}{E_{tot}}, \quad (2)$$

which was calculated as 28.5 %. While this method cannot precisely determine the amount of absorbance inside the film, it does nevertheless give a good estimation. This was confirmed by finite element modelling, where realistic temperatures were calculated inside the sample when this value was used as an input (see Figure 1 in the main article).

We used the UV/Vis spectrophotometer data for comparison only. Following reviewer’s advice, we calculated a weighted average of the absorbance. We obtained weight factors using the 800 V emission spectrum provided by PulseForge (Novacentrix) on their website (<https://pulseforge.com/frequently-asked-questions/>), see Figure 3 below. The sum of weight factors was one, and we used the spectrum in the range between 250 and 900 nm, for which we have the experimental absorbance values obtained with UV/Vis spectrophotometer.

Figure 3: **Emission spectrum of the xenon lamp** (from <https://pulseforge.com/frequently-asked-questions/>).

Thus calculated weighted absorption value is 2.8 %, which is an order of magnitude smaller compared to the bolometer value (28.5 %). This is physically unrealistic, as it gives too low temperatures when it is used in modelling (T_{\max} below 120 °C). There could be several reasons for this discrepancy:

- *Inaccurate spectrum data.* As can be observed in the Figure above and is also well reported in the literature ¹⁶, the position of the maximum in emission spectrum depends strongly on the applied voltage and pulse duration (the maximum shifts to lower wavelength values with increasing voltage). In our experiment we used ~890 V and 130 μ s for voltage and pulse length, respectively, which is different from the spectrum given above (800 V, 200 μ s). As the absorption in our sample appears mainly in the UV-range (see Figure 4), even a minor shift in the spectrum maximum can significantly increase the absorbance. As an example, if we had used the spectrum reported by Luo et al. ¹⁷ from Tulane University and NovaCentrix (750 V, 1.93 ms), we would obtain an average weighted absorbance of 12.0 %.
- *Sample transformation.* One must also consider the fact that the UV-Vis result probed pristine sample and the measurement does not have an influence on it. During the FLA experiment, as soon as the temperature exceeds pyrolysis temperature (350 °C), the sample starts to transform through several parallel processes (densification, removal of organic residues, crystallization, and grain growth). This significantly changes the optical properties of the sample, and these changes are not probed in the UV/Vis spectrophotometry.

Figure 4: **Absorbance of pyrolyzed PZT film on fused silica glass and Xe lamp emission spectrum.** The absorbance (blue curve) is calculated from transmittance and reflectance measure with a UV/Vis spectrophotometer. The Xe lamp spectra are obtained from Novacentrix (from <https://pulseforge.com/frequently-asked-questions/>) and Luo et al. ¹⁷ for applied voltages of 800 V and 750V and pulse durations of 200 μ s and 1.93 ms, respectively.

Following this discussion, we decided to remove the UV/Vis spectrophotometry data from the Supplementary Information as they can be misleading. We also modified the Supplementary to justify why bolometer measurements were used, as follow:

“To estimate the absorbance inside amorphous PZT films, a method using a bolometer was utilized ². First, the energy density on the surface of the stage of the flash-lamp annealer was measured with a bolometer after the light passed through air (direct exposure to light, $E_{tot} = 2.8 \text{ J cm}^{-2}$), bare fused silica glass (estimation of light reflected from the glass substrate assuming negligible absorption, $E_{glass} = E_{tot} - E_{ref} = 2.6 \text{ J cm}^{-2}$), and a fused silica glass with amorphous PZT on top (estimation of light that passes through the stack, $E_{sample} = E_{trans} = 1.8 \text{ J cm}^{-2}$), see Figure S1. To estimate the quantity of absorbed light, we used the equation:

$$E_{abs} = E_{tot} - E_{sample} - E_{glass} = E_{tot} - E_{trans} - (E_{tot} - E_{ref}). \quad (1)$$

In equation (1), E denotes energy density in J cm^{-2} . The resulting absorbed energy E_{abs} is 0.8 J cm^{-2} . The corresponding absorption value A is estimated as:

$$A = \frac{E_{abs}}{E_{tot}}, \quad (2)$$

which was calculated as 28.5 %.”

Note that this has no influence on the main paper. We have updated Figure S1 of the Supplementary as show in Figure 5.

Figure 5: **Transmitted energy density** measured with a bolometer after the light passing through: air (empty chamber), 500 μm -thick fused silica glass, and pyrolyzed PZT/fused silica stack. The energy density and the length of the applied light pulse were 2.8 J cm^{-2} and $130 \mu\text{s}$.

Comment 4.3

Please check the numbers in chapter 4.1. of the supplemental; for both high and low power densities an energy density of 2.5 J cm^{-2} is given. Furthermore, I doubt that you can separate nucleation and crystal growth so sharply, after exactly 6 pulses. It would be better to speak about 2 phases where nucleation or crystal growth is dominating.

Response 4.3

We confirm that, in the 2-step process, the energy density is of 2.5 J cm^{-2} for both the high and low power densities steps. What changes in the two cases is power density. It corresponds to the energy density per pulse divided by the pulse duration. In the 1st high power density step, the pulse duration is of $170 \mu\text{s}$ and it is increased to $250 \mu\text{s}$ in the 2nd low power density step of the process. This leads to a decrease of the power density from 14.7 kW cm^{-2} in the 1st step to 10.0 kW cm^{-2} in the 2nd step. We added the power density values in Section 2.5 of the main manuscript: “In the first step, 6 pulses with higher power density of 14.7 kW cm^{-2} (2.5 J cm^{-2} and $170 \mu\text{s}$) are applied. In the second step, 240 pulses with a lower power density of 10.0 kW cm^{-2} (2.5 J cm^{-2} and $250 \mu\text{s}$) are applied to grow the film at a lower temperature, thereby preventing the occurrence of cracks.”

We agree with the reviewer statement that the nucleation and the growth phases cannot be sharply separated. We changed the text in both the manuscript and the Supplementary Information to indicate that there are two phases where either nucleation or crystal growth is dominating:

- In Section 3 of the main manuscript: “The previously unreported process of two phases where either nucleation or grain growth of the perovskite phase is dominating”.
- In Section 4.1 of the Supplementary Information: “To address this issue, we have developed a two-step process consisting of stages where either nucleation or growth is dominating.”
- In Section 4.1 of the Supplementary Information: “In the second step, the phase where crystal growth is dominating”.

Reviewer 5

This manuscript entitled ‘Crystallization of piezoceramic films on glass via flash lamp annealing’ reports a flash lamp crystallization of PZT film on glass substrate. The annealed PZT film has a reasonable ferroelectric, piezoelectric, and transparent properties for various applications. Especially, the flash lamp annealing of solution-based PZT film on glass seems to have a novelty because the initial nature of solution coated PZT film is totally different with the physically coated films by sputtering and aerosol deposition. Additionally, the authors have effectively responded to the reviewers’ comments through the revision process. The revised manuscript could be accepted in Nature Communication with current form.

Response

We thank the reviewer for their very positive feedback.

References:

1. Bretos, I. *et al.* Activated solutions enabling low-temperature processing of functional ferroelectric oxides for flexible electronics. *Adv. Mater.* **26**, 1405–1409 (2014).
2. Tue, P. T., Shimura, R., Shimoda, T. & Takamura, Y. Direct integration of piezoactuator array with active-matrix oxide thin-film transistors using a low-temperature solution process. *Sensors Actuators, A Phys.* **295**, 125–132 (2019).
3. Ueda, K., Kweon, S. H., Hida, H., Mukouyama, Y. & Kanno, I. Transparent piezoelectric thin-film devices: Pb(Zr, Ti)O₃ thin films on glass substrates. *Sensors Actuators, A Phys.* **327**, 112786 (2021).
4. Greer, J. Large-area commercial pulsed laser deposition. in *Pulsed laser deposition of thin films* (ed. Eason, R.) 191–213 (John Wiley & Sons, Inc., 2007). doi:10.1002/9780470052129.ch9.
5. Schatz, A., Pantel, D. & Hanemann, T. Towards low-temperature deposition of piezoelectric Pb(Zr,Ti)O₃: Influence of pressure and temperature on the properties of pulsed laser deposited Pb(Zr,Ti)O₃. *Thin Solid Films* **636**, 680–687 (2017).
6. Park, K. Il *et al.* Highly-efficient, flexible piezoelectric PZT thin film nanogenerator on plastic substrates. *Adv. Mater.* **26**, 2514–2520 (2014).
7. Liu, T., Wallace, M., Trolier-McKinstry, S. & Jackson, T. N. High-temperature crystallized thin-film PZT on thin polyimide substrates. *J. Appl. Phys.* **122**, (2017).
8. Kim, J. H., Kweon, S. H. & Nahm, S. Low-temperature crystalline lead-free piezoelectric thin films grown on 2D perovskite nanosheet for flexible electronic device applications. *Nano Res.* **12**, 2559–2567 (2019).
9. Scott, J. F. Ferroelectrics go bananas. *J. Phys. Condens. Matter* **20**, (2008).
10. Mertin, S. *et al.* Piezoelectric and structural properties of c-axis textured aluminium scandium nitride thin films up to high scandium content. *Surf. Coatings Technol.* **343**, 2–6 (2018).
11. Godard, N. *et al.* 1-mW Vibration Energy Harvester Based on a Cantilever with Printed Polymer Multilayers. *Cell Reports Phys. Sci.* **1**, (2020).

12. Yao, Y. *et al.* Direct processing of PbZr_{0.53}Ti_{0.47}O₃ films on glass and polymeric substrates. *J. Eur. Ceram. Soc.* **40**, 5369–5375 (2020).
13. Palneedi, H. *et al.* Intense pulsed light thermal treatment of Pb(Zr,Ti)O₃/metglas heterostructured films resulting in extreme magnetoelectric coupling of over 20 V cm⁻¹ O-1. *Adv. Mater.* 2303553 (2023) doi:10.1002/adma.202303553.
14. Ouyang, J., Cormier, D., Williams, S. A. & Borkholder, D. A. Photonic sintering of aerosol jet printed lead zirconate titanate (PZT) thick films. *J. Am. Ceram. Soc.* **99**, 2569–2577 (2016).
15. Piper, R. T., Daunis, T. B., Xu, W., Schroder, K. A. & Hsu, J. W. P. Photonic Curing of Nickel Oxide Transport Layer and Perovskite Active Layer for Flexible Perovskite Solar Cells: A Path Towards High-Throughput Manufacturing. *Front. Energy Res.* **9**, 1–12 (2021).
16. Rebohle, L., Prucnal, S. & Reichel, D. *Flash lamp Annealing: From basics to applications*. (Springer International Publishing, 2019). doi:10.1007/978-3-030-23299-3.
17. Luo, S. *et al.* Instantaneous photoinitiated synthesis and rapid pulsed photothermal treatment of three-dimensional nanostructured TiO₂ thin films through pulsed light irradiation. *J. Mater. Res.* **32**, 1701–1709 (2017).

REVIEWER COMMENTS

Reviewer #3 (Remarks to the Author):

The authors have further improved the manuscript by removing speculative claims and adding useful clarifications. While the concern of the missing novelty can still be debated (it is a priority of the Editorial), I do not have any further remarks before accepting to publication. Merry Christmas!

Reviewer #4 (Remarks to the Author):

The authors comprehensively reviewed the manuscript, and I agree with the revisions according to comment 1 and 3. However, I still think that 28.5% absorbance is an overestimation. The authors explained their measurements in more detail, but nevertheless, in the bolometer method, it is assumed that the reflected part is in the order of 8% ($\sim 0.2 \text{ Jcm}^{-2}$) as expected for a quartz plate. This is nicely shown in Fig. S1 (or S1a formerly). However, if R and T measurements are correct, reflectance in the relevant UV part is at least 20 % (formerly Fig. S1b). Taking this as an approximation or lower limit, the reflected part is about 0.56 Jcm^{-2} , and the absorbance calculated by this method is in the order of 15.7 %, but immediately changes if I postulate a reflectance of e.g. 18 or 22%. But in any case 8% is too less. I'm also aware that this has a corresponding impact on the simulated temperatures.

I also understand now the problems with identifying the true Xe lamp spectrum. It is not even needed to have a strong change in the spectrum, a small shift of the falling UV edge of the spectrum is sufficient to have a large impact on the result if the sample is only absorbing in the UV. (I assume the weighted absorption was done at the energy scale, as otherwise the absorbance value give you the percentage of absorbed photons, not percentage of absorbed energy)

I also do not know the best solution for this. The easiest way is to keep the data but remark that 28.5% is a rough approximation, but the best guess you have. In this case, temperature profiles are a more general visualization of the profile. Alternately, someone has to check all optical measurements for hidden systematic deviations. One simple crosscheck could be: if the absorbance curve is right, the PZT sample should be transparent for the naked eye except a very small haze caused by the minimal absorbance between 400 and 450 nm.

I would agree with both methods of revision, and in this case I do not need to check the manuscript again.

Crystallization of piezoceramic films on glass via flash lamp annealing

Submission ID: NCOMMS-23-20861C

We thank all the reviewers for additional valuable comments. We considered them and implemented changes in the manuscript accordingly (highlighted in yellow). The comments are addressed in detail hereafter, point by point.

Reviewer 3

The authors have further improved the manuscript by removing speculative claims and adding useful clarifications. While the concern of the missing novelty can still be debated (it is a priority of the Editorial), I do not have any further remarks before accepting to publication. Merry Christmas!

Response:

We thank the reviewer for their positive feedback and for their time evaluating the manuscript.

Reviewer 4

The authors comprehensively reviewed the manuscript, and I agree with the revisions according to comment 1 and 3.

Response

We thank the reviewer for their positive feedback.

Comment 4.1

However, I still think that 28.5% absorbance is an overestimation. The authors explained their measurements in more detail, but nevertheless, in the bolometer method, it is assumed that the reflected part is in the order of 8% ($\sim 0.2 \text{ Jcm}^{-2}$) as expected for a quartz plate. This is nicely shown in Fig. S1 (or S1a formerly). However, if R and T measurements are correct, reflectance in the relevant UV part is at least 20 % (formerly Fig. S1b). Taking this as an approximation or lower limit, the reflected part is about 0.56 Jcm^{-2} , and the absorbance calculated by this method is in the order of 15.7 %, but immediately changes if I postulate a reflectance of e.g. 18 or 22%. But in any case 8% is too less. I'm also aware that this has a corresponding impact on the simulated temperatures.

Response 4.1

We do agree with the reviewer that bolometer measurements (Fig. S1) give rough estimation of the absorbance only, however, we do believe that they give values that are more realistic for photonic annealing process than the values obtained with UV/Vis spectrophotometry (former Fig. S1b). The main reason is transformation of the amorphous sample upon processing (densification, removal of organic residues, crystallization, and grain growth), which does not happen during UV/Vis measurements. This can also explain large discrepancies between the two experiments.

Comment 4.2

I also understand now the problems with identifying the true Xe lamp spectrum. It is not even needed to have a strong change in the spectrum, a small shift of the falling UV edge of the spectrum is sufficient to have a large impact on the result if the sample is only absorbing in the UV. (I assume the weighted absorption was done at the energy scale, as otherwise the absorbance value give you the percentage of absorbed photons, not percentage of absorbed energy).

Response 4.2

We confirm that the weighted absorption was done at the energy scale.

Comment 4.3

I also do not know the best solution for this. The easiest way is to keep the data but remark that 28.5% is a rough approximation, but the best guess you have. In this case, temperature profiles are a more general visualization of the profile. Alternately, someone has to check all optical measurements for hidden systematic deviations. One simple crosscheck could be: if the absorbance curve is right, the PZT sample should be transparent for the naked eye except a very small haze caused by the minimal absorbance between 400 and 450 nm.

I would agree with both methods of revision, and in this case I do not need to check the manuscript again.

Response 4.3

Please see the appearance of a spin-coated sample (after pyrolysis, before flash-lamp annealing) in Figure 1. The layer is transparent with small haze and color due to absorption/reflection appearing near 400 nm-range, as expected from UV/Vis experiment (former Fig. S1b). Its appearance changes upon flash lamp annealing because the sample transforms during its first exposure to the light pulse.

Figure 1: **Appearance of a PZT film after pyrolysis (before flash lamp annealing)**. Circular edge of the sample is coloured differently due to higher thickness forming upon spin coating.

The reviewer gives us two options – to keep the bolometer data (and mention in the manuscript that this gives rough estimation of absorption) or to revert to optical data. We opted for the first option as the experimental conditions during bolometer measurements are closer to the flash lamp annealing process.

To follow reviewer's advice, we also modified the manuscript:

- We added a sentence in the caption of Figure 1: “Absolute temperature values are indicative only.” The fact that bolometer measurements are approximations has already been mentioned in the main text of the previous version, see the sentence in Section 2.1: “The absorbance of an amorphous PZT layer (~28.5 %) was estimated with a bolometer placed below the sample, [...]”
- We added two sentences in the Supplementary, Section 1: “While this method is not ideal, it is a reasonable approach as the measurement is performed under the same conditions as flash lamp annealing.” and “Note that this value is a rough estimation only.”

Additional Corrections

We noticed a minor typographic error in our previous answer to reviewer 4. In Equation (1) of Supplementary:

$$E_{abs} = E_{tot} - E_{sample} - E_{glass} = E_{tot} - E_{trans} - (E_{tot} - E_{ref}),$$

the indexes of E_{glass} and E_{ref} were inverted. Correct version now reads as:

$$E_{abs} = E_{tot} - E_{sample} - E_{ref} = E_{tot} - E_{trans} - (E_{tot} - E_{glass}) \quad (1)$$

Note that this does not change the calculation results.